# JIP4 and RILPL1 utilize opposing motor force to dynamically regulate lysosomal tubulation

Luis Bonet-Ponce[1,2,3], Tsion Tegicho[1,2], Nuria Fernandez-Martinez[1,2], Irene A. Rozenberg[1,2], Mia Ashriem[1,2], Alexandra Beilina[3], Jillian H. Kluss[3], Yan Li[4], and Mark R. Cookson[3]

**Lysosomes are dynamic organelles that remodel their membrane in response to stimuli. We previously uncovered a process we term LYsosomal Tubulation/sorting driven by LRRK2 (LYTL), wherein damaged lysosomes generate tubules sorted into vesicles. LYTL is orchestrated by the Parkinson's disease kinase LRRK2 that recruits the motor adaptor protein and RHD family member JIP4 to lysosomes. JIP4 enhances LYTL tubule extension toward the plus-end of microtubules. To identify new players involved in LYTL, we mapped the lysosomal proteome after LRRK2 kinase inhibition. We found that RILPL1 is recruited to dysfunctional lysosomes in an LRRK2 kinase activity–dependent manner, facilitated by pRAB proteins. Unlike JIP4, RILPL1 induces retraction of LYTL tubules by binding to p150[Glued], thereby moving lysosomal tubules toward the minus-end of microtubules. Our findings emphasize the dynamic regulation of LYTL tubules by two distinct RHD proteins and pRAB effectors, acting as opposing motor adaptor proteins. These opposing forces create a metastable lysosomal membrane deformation, enabling dynamic tubulation events.**

## Introduction

Lysosomes were initially discovered by Christian de Duve (de Duve et al., 1955) as acidic organelles responsible for the hydrolysis of macromolecules. Beyond cargo clearance, lysosomes play a multifaceted role within the cell (Ballabio and Bonifacino, 2020). Lysosomes control nutrient responses (Goul et al., 2023), regulate the morphology of other organelles (Spits et al., 2021; Wong et al., 2018), facilitate cholesterol egress (Chu et al., 2021; Höglinger et al., 2019), and co-transport other cellular structures (Liao et al., 2019; Guo et al., 2018). Thus, lysosomes serve as central organelles that help maintain cellular homeostasis, particularly during periods of cellular stress (Tan and Finkel, 2023).

When the integrity of the lysosomal membrane is compromised, cells can initiate "lysosomal cell death" (Wang et al., 2018). This process involves the leakage of protons and hydrolases into the cytosol, ultimately damaging multiple cellular macromolecules and terminating cell viability. We and others have characterized different pathways by which lysosomes deal with membrane damage. When the damage is limited, lysosomes can repair their membrane through ESCRT (Skowyra et al., 2018; Radulovic et al., 2018; Herbst et al., 2020), lysosomal sphingomyelin scrambling (Niekamp et al., 2022), or lipid exchange with the ER (Tan and Finkel, 2022; Radulovic et al., 2022). RNA

granules can also plug rupture sites to prevent lysosomes from collapsing and enable an efficient repair response (Bussi et al., 2023). When damage persists, cells clear ruptured lysosomes via selective autophagy in a process known as lysophagy (Maejima et al., 2013). Additionally, lysosomal membrane damage can induce a noncanonical autophagy pathway called conjugation of ATG8 to single membranes (CASM) (Kaur et al., 2023; Boyle et al., 2023; Corkery et al., 2023), which might be associated with lysosomal membrane repair (Cross et al., 2023). Collectively, the existence of these multiple, partially redundant pathways indicates how critical it is for cells to avoid lysosomal membrane damage.

Beyond membrane repair and autophagy, lysosomal membrane damage triggers a unique tubulation and membrane sorting process, which we have named LYsosomal Tubulation/sorting driven by leucine-rich repeat kinase 2 (LRRK2) (LYTL) (Bonet-Ponce et al., 2020; Bonet-Ponce and Cookson, 2022b). LYTL is orchestrated by LRRK2, a large kinase basally located in the cytosol that is activated at membranes (Bonet-Ponce and Cookson, 2022a), where it phosphorylates a subset of RAB GTPases (Steger et al., 2016; Kluss et al., 2022b). Recent evidence indicates that LRRK2 is recruited to ruptured lysosomes

[1]Department of Neurology, The Neuroscience Research Institute, College of Medicine, The Ohio State University Wexner Medical Center, Columbus, OH, USA; [2]Department of Biological Chemistry and Pharmacology, The Ohio State University, Columbus, OH, USA; [3]Cell Biology and Gene Expression Section, National Institute on Aging, National Institutes of Health, Bethesda, MD, USA; [4]Proteomic Core Facility, National Institute of Neurological Disorders and Stroke, National Institutes of Health, Bethesda, MD, USA.

Correspondence to Luis Bonet-Ponce: luis.bonetponce@osumc.edu; Mark R. Cookson: cookson@mail.nih.gov.

via RAB12 (Wang et al., 2023; Dhekne et al., 2023) and the CASM machinery (Bentley-DeSousa and Ferguson, 2023, *Preprint*; Eguchi et al., 2024), where it phosphorylates and brings pRAB proteins to the lysosomal membrane (Eguchi et al., 2018; Herbst et al., 2020; Bonet-Ponce et al., 2020). Subsequently, pRAB10 recruits its effector JIP4 C-jun-amino-terminal kinase–interacting protein 4 (JIP4, a motor adaptor protein) that promotes the formation of a lysosomal tubule negative for lysosomal membrane markers (Bonet-Ponce et al., 2020; Bonet-Ponce and Cookson, 2022b; Kluss et al., 2022a). The JIP4-positive LYTL tubule can be resolved into vesicles that travel through the cytosol and contact healthy lysosomes. LYTL is regulated by lysosomal positioning (Kluss et al., 2022a), and the membrane sorting step is controlled by the ER (Bonet-Ponce and Cookson, 2022b). However, the entirety of molecular pathways that regulate LYTL remain undefined.

Here, we use unbiased proteomics on isolated lysosomes to characterize the repertoire of proteins whose abundance on lysosomes depends on the activity of LRRK2. We show that LRRK2 recruits RILP-like protein 1 (RILPL1) to ruptured lysosomes in a pRAB-dependent manner. RILPL1 is a RILP homology domain (RHD) family member that clusters LRRK2-positive lysosomes to the perinuclear area and retracts LYTL tubules. Mechanistically, RILPL1 binds to p150$^{Glued}$ (a dynactin subunit), enhancing its recruitment to the damaged lysosomes and promoting the transport of lysosomes and tubules to the minus-end of microtubules. By further characterizing the tubulation process, we demonstrate that LYTL tubules move along tyrosinated microtubules, and tubulin tyrosination is required for tubule elongation. In summary, our findings underscore the dynamic regulation of LYTL tubules by two distinct pRAB effectors, functioning as opposing motor adaptor proteins: JIP4, which promotes tubulation, and RILPL1, which facilitates tubule retraction. Collectively, these two opposing forces lead to a metastable membrane structure that supports a dynamic and transient tubulation event.

## Results

### Using unbiased proteomics to uncover the LRRK2 lysosomal proteome

As we have previously established, LRRK2 is a cytosolic protein that is recruited to the lysosomal membrane in the presence of the lysosomotropic damaging agent L-leucyl-L-leucine methyl ester (LLOME) (Fig. 1, A–C) (Bonet-Ponce et al., 2020; Bonet-Ponce and Cookson, 2022b; Kluss et al., 2022a). LRRK2-positive lysosomes are likely to have damaged membranes, as ~66% of them colocalize with the damage marker galectin-3 (Fig. 1, D and E). However, only ~14% of the galectin-3–positive lysosomes are stained with LRRK2, suggesting that galectin-3 and LRRK2 have different recruitment mechanisms and may play a different role in maintenance of lysosome and cellular homeostasis. To establish a comprehensive map of the LRRK2 lysosomal proteome, we created a HEK293T cell line that stably expresses GFP-LRRK2 and TMEM192-3xHA. TMEM192 is a lysosomal membrane protein used as bait to immuno-isolate lysosomes in a technique known as lysosome

immunoprecipitation (LYSO-IP) (Abu-Remaileh et al., 2017). We confirmed that, in the presence of the lysosomotropic damaging agent LLOME, LRRK2 colocalized with TMEM192 (Fig. S1, A and B). We also confirmed the presence of endogenous RAB10 and JIP4 on lysosomes, which were completely abrogated by LRRK2 kinase inhibition using MLi2 (Fig. S1, C and D). We next validated the LYSO-IP as a suitable technique to isolate lysosomes (Fig. S1, E–G) and to capture the enrichment of pRAB10, total RAB10, and JIP4 to the lysosomal fraction in an LRRK2 kinase–dependent manner (Fig. S1 F).

We reasoned that any proteins recruited by LRRK2 to lysosomes would be present on isolated lysosomes from LLOME-treated cells but absent in cells pre-treated with MLi2 (Fig. 2, A and B). To validate that we could detect changes in the lysosomal proteome, we compared DMSO- and LLOME-treated cells. As expected, LLOME addition promoted the recruitment of members of the ESCRT complex, known autophagy/lysophagy proteins (Eapen et al., 2021), and LRRK2. We also observed a significant decrease in lysosomal hydrolases upon LLOME treatment (Fig. S1, I and J). Notably LRRK2 kinase inhibition did not modify the lysosomal presence of any of the ESCRT members, autophagy proteins, or lysosomal hydrolases analyzed (Fig. S1, I and J), suggesting that LRRK2 is not involved in maintaining the integrity of the lysosomal membrane nor promoting lysosomal exocytosis.

We also found that 46 proteins were significantly increased on lysosomes from cells treated with LLOME compared with co-treatment with LLOME and MLi2 (Fig. 2 C). The 46 candidates were enriched for membrane trafficking terms (Fig. 2 D), including "vesicle docking involved in exocytosis," "protein secretion," "response to calcium," "protein localization to plasma membrane," and "establishment of vesicle localization." Specifically, nine RAB GTPases were identified in our screen (RAB3A/B/D, RAB8A/B, RAB10, RAB12, RAB29, and RAB35) (Fig. 2, C, E, and F; and Fig. S2 A). All these RABs have been proposed as LRRK2 substrates (Steger et al., 2016), and no other RAB protein was identified. Interestingly, RAB5A/B/C and RAB43 (Fig. 2 F and Fig. S2 A) were also originally proposed as LRRK2 substrates but were not recovered in our survey, suggesting that these RABs are not *bona fide* substrates in the context of lysosomal damage. We also identified eight calcium-binding proteins, six RNA-binding proteins, and eight dynein-associated proteins (Fig. 2 E). Consistent with our previous work, we detected JIP4 as a positive hit, further validating our screening strategy (Fig. 2, C, E, and F; and Fig. S2 B). RILPL1 was detected as a new LRRK2-interacting candidate on lysosomes (Fig. 2, C, E, and F; Fig. S1 K; and Fig. S2 B). RILPL1 is a RHD family member, along with RILPL2, JIP3, and JIP4. All four proteins are predicted to act as pRAB effectors (Waschbüsch et al., 2020). RILPL1 has been previously linked to LRRK2 and has been reported to reduce ciliogenesis and alter centrosomal cohesion (Dhekne et al., 2018; Lara Ordóñez et al., 2019). Therefore, we further examined RILPL1 as a potential mediator of responses to lysosomal membrane damage.

### LRRK2 recruits RILPL1 to lysosomes via pRAB proteins

We used immunostaining as an orthogonal approach to validate that the recruitment of RILPL1 to lysosomes occurs across cell

Figure 1. **LRRK2 recruitment to membrane damaged lysosomes. (A)** U2OS cells were transfected with HaloTag-LRRK2 for 36 h and then imaged live after treatment with DMSO or LLOME (2 h). **(B)** U2OS cells transfected with HaloTag-LRRK2 and LAMP1-RFP for 36 h were treated with LLOME (2 h) and imaged live. **(C)** 3D view using "volume view" plugin from Fiji. **(D)** U2OS cells were transfected with HaloTag-LRRK2 and GFP-GAL3 for 36 h and treated with LLOME. **(E)** Graph shows the percentage of total LRRK2⁺ lysosomes that colocalize with GAL3 (yellow) and the percentage of total GAL3⁺ lysosomes that colocalize with LRRK2 (cyan) per cell. Data are mean ± SEM. *n* = 21 cells. Scale bar A = 10 µm; (B–E) = 1 µm. Insets (A) = 2 µm.

types in a manner that depends on LRRK2. Endogenous RILPL1 was detected on LRRK2-positive lysosomes after LLOME treatment in HEK293T cells, U2OS cells, and mouse primary astrocytes (Fig. S3, A and B; and Fig. 3 A). As expected, we could not detect RILPL1 presence on lysosomes after LRRK2 kinase inhibition (Fig. S3 A; and Fig. 3, A and B). The anti-RILPL1 antibody used in this study was validated against siRNA for western blot (WB) and immunocytochemistry (ICC) (Fig. S3, C and D). RILPL1 recruitment to LRRK2-positive lysosomes was also observed in live cells (Fig. 3 C) using exogenous expression of mCherry-RILPL1 and LAMP1-mNeonGreen. The presence of active LRRK2 on lysosomes is sufficient to recruit RILPL1, as we observed RILPL1 presence on lysosomes when trapping LRRK2 to the lysosomal membrane using a previously described chimera construct expressing the first 39 amino acids of LAMTOR1 (LYSO-LRRK2 plasmid) (Kluss et al., 2022a) (Fig. S3 E), thereby bypassing the need to damage lysosomal membranes using LLOME. We hypothesized that RILPL1 is recruited to LRRK2-positive lysosomes via pRAB proteins. Supporting this contention, we observed a physical interaction through immunoprecipitation between a tagged version of RILPL1 and pRAB10 in LLOME-treated cells (Fig. 3 D). This interaction was further reproduced in U2OS cells and was, again, dependent on LRRK2 kinase activity (Fig. S3 F). Interestingly, we observed a strong interaction between RILPL1 and pRAB12 in control conditions that was exacerbated in LLOME-treated cells (Fig. S3 G).

JIP3, JIP4, RILPL1, and RILPL2 have two conserved arginine (R) residues in their RILP homology domain 2 (RHD2) that are predicted to mediate the binding to the phosphorylated residue

of the pRAB substrates (Waschbüsch et al., 2020) (Fig. 3 E). Thus, this group of proteins is proposed to act as pRAB effectors. Previous *in vitro* structural biology studies have demonstrated that RILPL2 binds to RAB8A-pThr.72 through an exposed arginine (R132) in its RHD2 region (Waschbüsch et al., 2020). The binding is enhanced by a nearby Arg. residue (R130) (Waschbüsch et al., 2021). Given that both R residues are conserved in RILPL1, we considered that a similar binding mechanism might occur (Fig. 3 E). Our structural modeling using AlphaFold predicts that each monomer of an antiparallel RILPL1 RHD2 domain dimer interacts with pThr.73-RAB10 via Arg.293, forming a tetramer (Fig. 3 F). We focused on RAB10 for two main reasons: (1) it is the pRAB with the best characterized antibody, and (2) we previously showed that RAB10 binds and recruits JIP4 to lysosomes (Bonet-Ponce et al., 2020). When Arg.293 was mutated into alanine (RILPL1-R293A), LRRK2 was unable to recruit RILPL1 to the lysosomal membrane in cells treated with LLOME (Fig. 3 G). Furthermore, the expression of a phospho-null RAB10 mutant (T73A) strongly decreases the presence of endogenous RILPL1 on lysosomes (Fig. 3 H). We then tested if RILPL1 and JIP4 compete for interaction with pRABs. Surprisingly, we observed a stronger binding between RILPL1 and pRAB10 upon JIP4 overexpression (Fig. 3, I and J), suggesting a positive effect of JIP4 toward the pRAB10:RILPL1 complex. We did not detect a direct interaction between RILPL1 and JIP4 (Fig. S3 H), suggesting that a tripartite complex involving pRABs and both effectors is unlikely. Altogether, our data strongly suggest that RILPL1 is a pRAB effector, with a similar mode of binding to the previously described effector JIP4 but without competition.

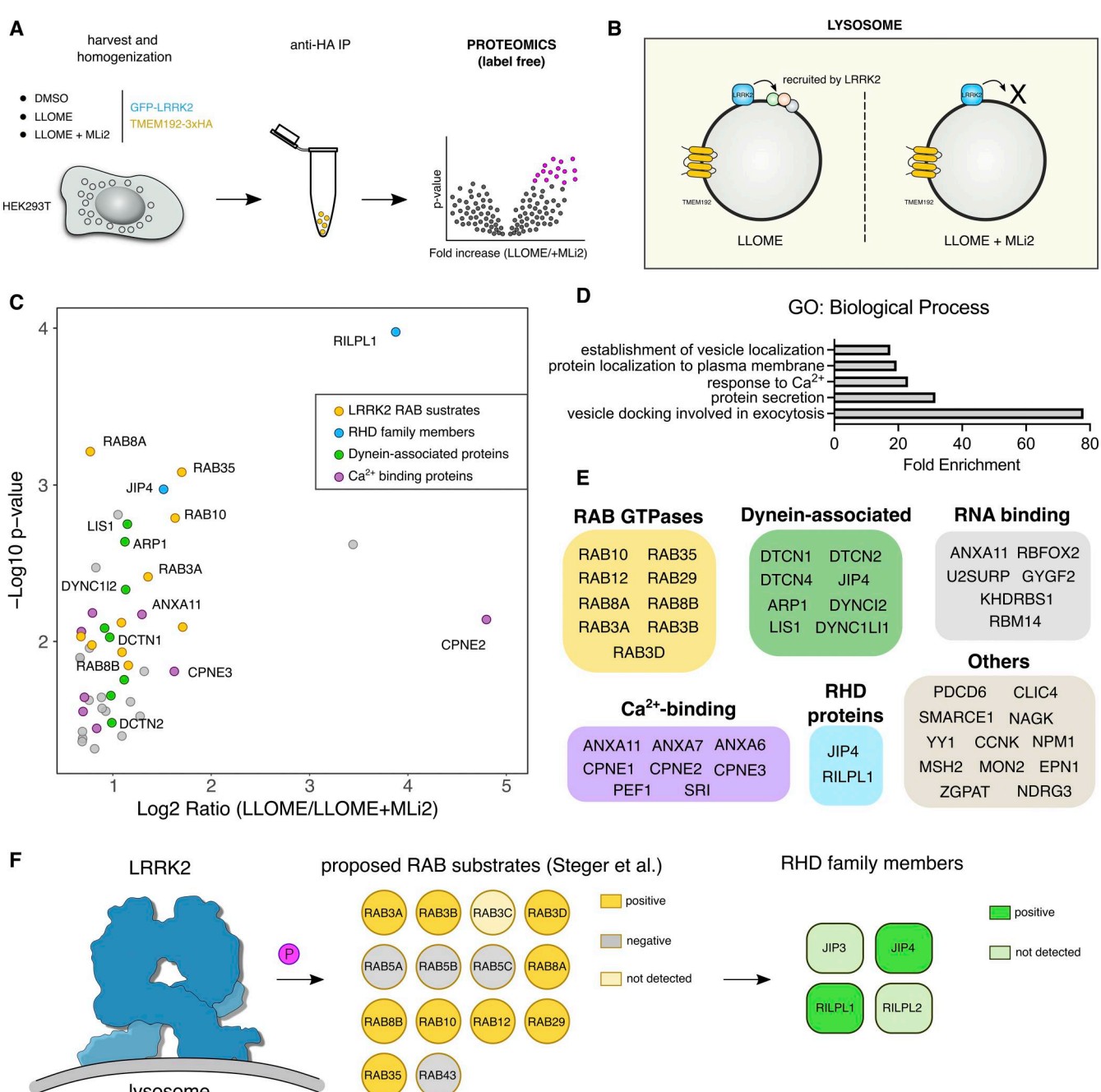

Figure 2. **Unbiased characterization of the LRRK2 lysosomal proteome. (A and B)** Schematic depiction of the experimental workflow of the lysosomal immunoprecipitation (LYSO-IP) in HEK293T stably expressing LRRK2, reasoning that the proteins recruited by LRRK2 to damaged lysosomes will be absent if the cells are pre-treated with the LRRK2 kinase inhibitor MLi2. **(C)** Volcano plot showing the subset of proteins with a fold change >1.3 and P value <0.05 present on the lysosomal fraction of cells treated with LLOME vs LLOME+MLi-2. Data are from four independent experiments. **(D)** Gene Ontology (GO) search of the top five enriched terms for Biological Process of the 46 candidates found in C, with fold enrichment indicated on the horizontal axis. **(E)** The 46 candidate proteins were grouped by molecular function. **(F)** Model of the LRRK2 lysosomal proteome, where LRRK2 recruits nine RAB substrates and two RHD family members to damaged lysosomes.

## RILPL1 clusters LRRK2-positive lysosomes to the centrosome and reduces LYTL tubulation

As we previously reported, LRRK2-positive lysosomes cluster around the centrosome upon LLOME treatment (Kluss et al., 2022a) (Fig. 4 A). Interestingly, in cells lacking RILPL1, the clustering of LRRK2-positive lysosomes around the centrosome was drastically reduced (Fig. 4, A and B) even though they are

still close to the nucleus (Fig. 4, A and C). We next wondered if LRRK2 recruitment is restricted to lysosomes clustered around the centrosome, or alternatively if LRRK2 recruitment precedes and promotes perinuclear clustering. Time-lapse microscopy experiments demonstrated that LRRK2 recruitment precedes the clustering of LRRK2-positive lysosomes in the perinuclear space (Fig. 4 D and Video 1) and is decreased in cells treated with

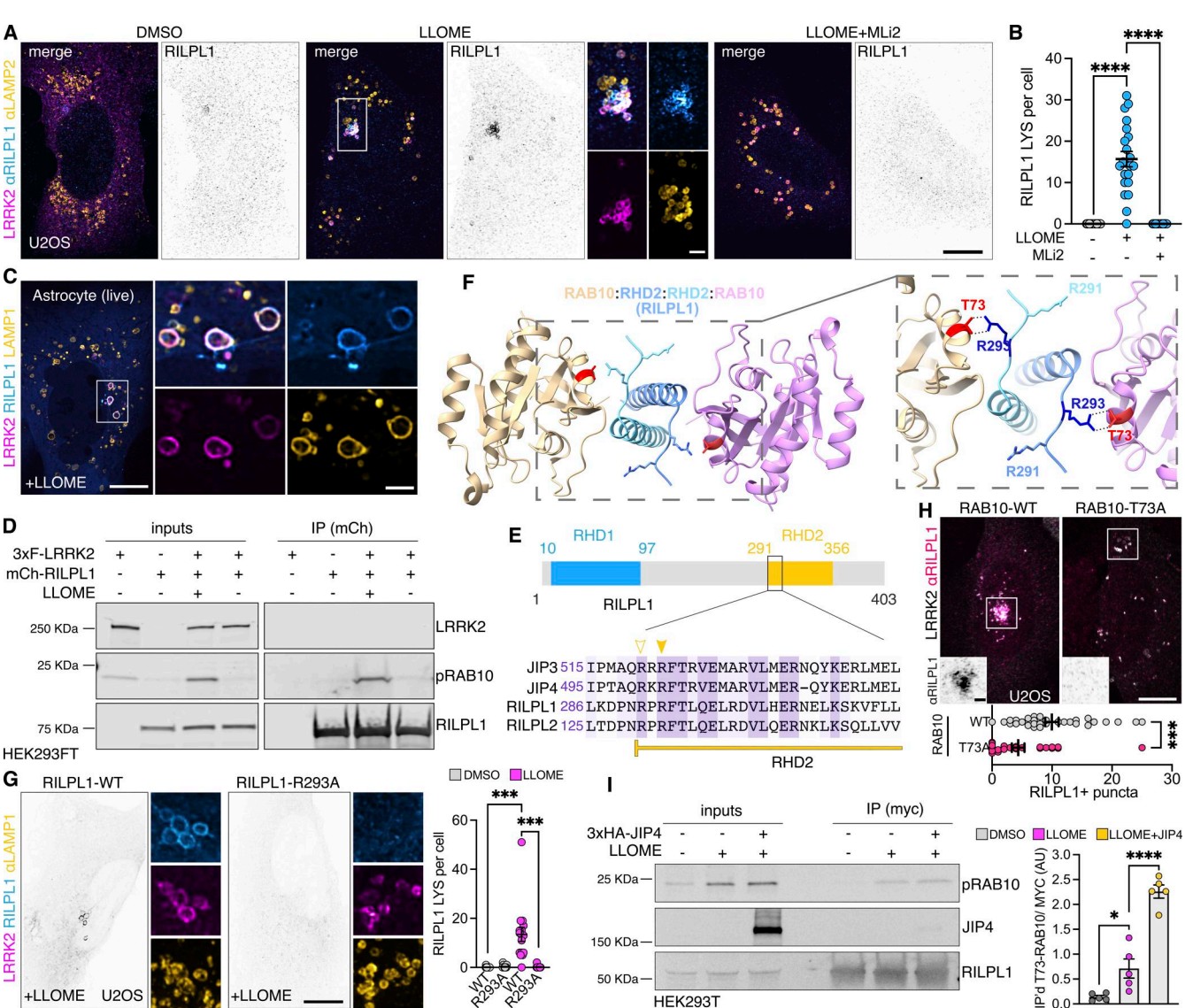

Figure 3. **LRRK2 recruits RILPL1 to damaged lysosomes via pRAB proteins. (A)** U2OS cells were transfected with HaloTag-LRRK2 for 36 h. Cells were then treated with DMSO, LLOME, or LLOME+MLi2 for 2 h before fixation. Cells were then stained for RILPL1 and LAMP2. **(B)** Quantification of the RILPL1 lysosomes per cell in the three different conditions. Data are mean ± SEM (P < 0.0001, n = 21 cells). One-way ANOVA with Tukey's post hoc tests. **(C)** Mouse primary astrocytes were transfected with HaloTag-LRRK2, mCherry-RILPL1, and LAMP1-mNeonGreen for 48 h. Astrocytes were treated with LLOME for 6 h and imaged live. **(D)** HEK293FT cells were transfected with 3xflag-LRRK2 and mCherry-RILPL1 for 24 h. Cells were treated or not with LLOME for 2 h, and lysates were subjected to immunoprecipitation with anti-RFP antibodies. RFP, LRRK2, and pT73-RAB10 were blotted. **(E)** RILP-RHD2 alignment of JIP3, JIP4, RILPL1, and RILPL2 using Clustal (Clustal Omega, EMBL). Arrowheads show the conserved Arg. residues responsible for the binding with the pRAB substrates. Filled arrowhead marks the Arg. that binds with the phosphorylated residue, and the empty arrowhead indicates the Arg. that stabilizes the interaction. **(F)** Structural model of the interaction between the RHD2 of RILPL1 and RAB10, showing the interaction between T73-RAB10 and R293-RILPL1. **(G)** U2OS cells were transfected with HaloTag-LRRK2 and mCherry-RILPL1 (WT or R293A) for 36 h. Cells were then treated with DMSO or LLOME for 2 h, fixed, and stained for LAMP1. Quantification of RILPL1-positive lysosomes per cell in the different groups. Data are mean ± SEM (P = 0.0009 and P = 0.001, n = 16 cells). Brown–Forsythe and Welch ANOVA with Dunnett's post hoc test. **(H)** U2OS cells were transfected with 3xflag-LRRK2, along with 2xmyc-RAB10 (WT) or 2xmyc-RAB10 (T73A). Cells were then treated with LLOME for 2 h, fixed, and stained with anti-flag, anti-myc, and anti-RILPL1 antibodies. Graph shows the quantification of RILPL1 puncta per cell in the different conditions. Data are mean ± SEM (P= 0.0006, n = 28–32 cells). Unpaired t test. **(I)** HEK239T cells were transfected with HaloTag-LRRK2 and 2xmyc-RILPL1, and one group was also transfected with Ubc-3xHA-JIP4 for 24 h. Cells were then treated or not with LLOME (2 h) and subjected to anti-myc immunoprecipitation. Graph shows the amount of RILPL1:T73-RAB10 interaction in the different groups. Data are mean ± SEM (P < 0.0001, P= 0.0293; N = 5 independent replicates). One-way ANOVA with Tukey's post hoc test. Scale bar (A, G, and H) = 10 μm; (C) = 20 μm. Insets (A and H) = 2 μm; (C) = 4 μm. Source data are available for this figure: SourceData F3.

RILPL1 siRNA (Fig. 4 E). In U2OS cells lacking exogenous LRRK2 expression, RILPL1 knockdown has no effect in lysosomal positioning in untreated cells or LLOME-treated cells (Fig. S3, I–K), suggesting that the role of RILPL1 on lysosomal positioning is

LRRK2-dependent. DYNC1H1 knockdown decreases the perinuclear clustering of LRRK2-positive lysosomes, even in cells that exogenously express RILPL1 (Fig. 4, F and G). Collectively, these data suggest that RILPL1 is responsible for clustering LRRK2-

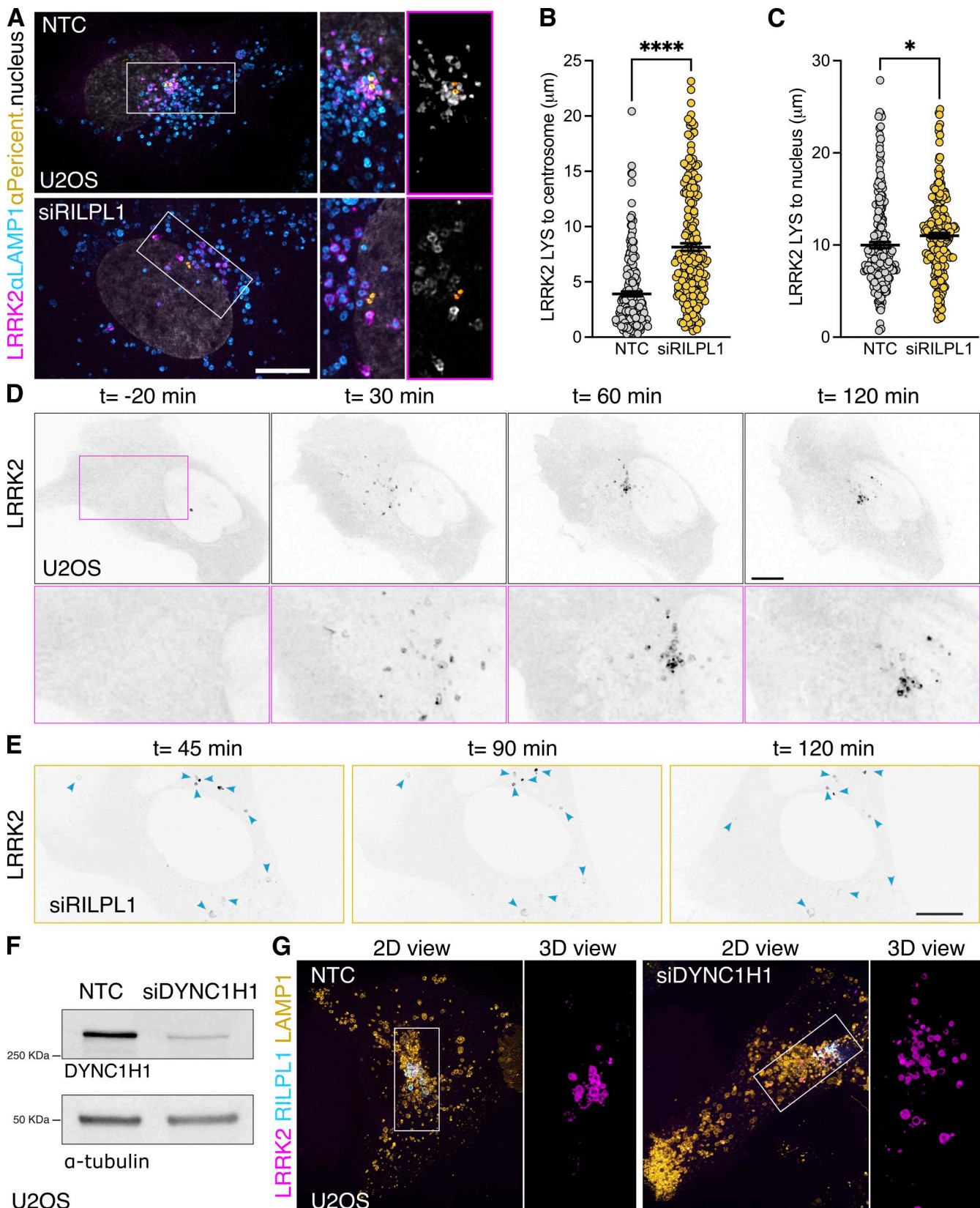

Figure 4. **RILPL1 promotes LRRK2+ lysosome clustering around the centrosome. (A)** U2OS cells were treated with a nontargeting control (NTC) or siRILPL1 RNA. 24 h later, cells were transfected with HaloTag-LRRK2 for 36 h. Then, cells were treated with LLOME (2 h) and were fixed and stained with a pericentrin antibody. Volume view image of LRRK2+ lysosomes clustered around the centrosome (pericentrin). **(B)** The distance between LRRK2+ lysosomes and the centrosome was measured in cells treated with NTC or siRILPL1. Data are mean ± SEM (P < 0.0001, n = 221–222 lysosomes, from 7 to 9 cells). Unpaired

*t* test with Welch's correction. **(C)** The distance between LRRK2⁺ lysosomes and the center of the nucleus was measured in cells treated with NTC or siRILPL1. Data are mean ± SEM (P = 0.0206, n = 221–222 lysosomes, from 7 to 9 cells). Unpaired *t* test. **(D and E)** Time-lapse of a U2OS cell expressing HaloTag-LRRK2 and treated with LLOME. LRRK2 puncta recruitment and movement are followed at different time points up to 120 min. **(D)** is a control cell, and (E) is a cell previously transfected with a RILPL1 siRNA. **(F)** U2OS cells treated with a NTC and siRNA against DYNC1H1 were lysed and blotted against anti-DYNC1H1 and a-tubulin antibodies. **(G)** U2OS cells were incubated with NTC or siDYNC1H1 (24 h) and transfected with HaloTag-LRRK2, mBaoJin-RILPL1, and LAMP1-RFP (36 h). Cells were then imaged live after treatment with LLOME (2 h). Arrowheads indicate LRRK2⁺ lysosomes. Scale bar = 10 μm. Insets = 2 μm. Source data are available for this figure: SourceData F4.

positive lysosomes around the centrosome. We have previously shown that perinuclear clustering promotes the ability of LRRK2 to recruit JIP4 and initiate LYTL (Kluss et al., 2022a). While RILPL1 knockdown does not affect LRRK2 recruitment to lysosomes (Fig. 5, A and B), it did significantly decrease the presence of JIP4 on LRRK2-positive lysosomes (Fig. 5, A and C–E). Our results are consistent with our prior observations (Kluss et al., 2022a) that perinuclear clustering of LRRK2-positive lysosomes is important to recruit JIP4.

Based on these results, we considered if RILPL1 might also play a role in LYTL. Super-resolution microscopy identified RILPL1 tubules that are negative for LRRK2 and LAMP1 but that emanate from LRRK2-positive lysosomes (Fig. 6 A). Additionally, RILPL1 colocalizes with JIP4 on the lysosomal tubules, demonstrating that both effectors bind the same LYTL-derived structures (Fig. 6 B). The presence of RILPL1 on LYTL tubules was corroborated at the endogenous level (Fig. S3 L) and is therefore not an artifactual consequence of overexpression. RILPL1 tubules were also able to undergo fission (Fig. 6 C), behaving similarly to JIP4-positive tubules (Video 2) (Bonet-Ponce and Cookson, 2022b). RILPL1 knockdown leads to a modest increase in tubule length (Fig. 6, D and E) but a strong increase in tubulation (Fig. 6, D and F), which measures the number of JIP4-positive tubules divided by the number of JIP4-positive lysosomes in each cell. These results are in contrast to the effect of JIP4, decreasing tubule length (Fig. 6 H) and tubulation (Fig. 6 I), in agreement with our previous work (Bonet-Ponce et al., 2020). We therefore infer that while both effector proteins bind pRABs at the LYTL tubules, JIP4 promotes tubule elongation and RILPL1 triggers tubule retraction, generating a dynamic metastable structure from the lysosomal surface. Consistently, RILPL1 depletion leads to a decrease in dynamic tubular events (Fig. 6, J and K).

### RILPL1 binds to p150^Glued and drives organelle movement to the minus-end of microtubules

A common feature of the RHD family members is their role as motor adaptor proteins. JIP3 and JIP4 can bind to dynein/dynactin and kinesins, while RILPL2 binds to myosin-Va. Although RILPL1 has not previously been associated with any motor, *in vitro* work has proposed a potential interaction between RILPL1 and dynein-light intermediate chain (DLIC) (Celestino et al., 2022). As dynein/dynactin moves organelles toward the centrosome, we asked whether RILPL1 mediates its function on LRRK2-positive lysosomes by acting as a dynein adaptor protein. This hypothesis would be consistent with the observation that LYTL tubules elongated toward the plus-end of the microtubules, away from the centrosome (Fig. 7, A–C), further suggesting that LYTL tubules elongate via kinesin and retract

through dynein/dynactin. Given that JIP3 and JIP4 bind to both DLIC and to the dynactin subunit p150^Glued (DCTN1) (Vilela et al., 2019), we assessed if RILPL1 is able to bind to these proteins. Immunoprecipitation assays reveal that RILPL1 binds to p150^Glued, although we could not reproduce the RILPL1:DLIC interaction in our experimental conditions (Fig. 7 D). RILPL1: p150^Glued binding is increased upon LLOME addition, supporting our model. Our mass spectrometry data on isolated lysosomes show an LRRK2 kinase–dependent recruitment of three cytoplasmic dynein subunits, the dynein activator LIS1, and four dynactin complex members, including p150^Glued (Fig. S2 C). We validated the recruitment of p150^Glued using western blotting (Fig. 7, E and F), demonstrating that LRRK2 kinase activity is required to recruit p150^Glued to damaged lysosomes in a similar pattern to RILPL1. We then reasoned that LRRK2 kinase activity recruits p150^Glued through RILPL1. RILPL1 knockdown significantly reduces the amount of p150^Glued on lysosomes in LLOME-treated cells (Fig. 7, G and H), suggesting that RILPL1 is partially responsible for the recruitment of dynactin to damaged lysosomes (Fig. 7 I). To address whether RILPL1 is sufficient to promote transport of organelles toward the nucleus, we used the FKBP–FRB system to localize RILPL1 to mitochondria by co-expressing a Mito-YFP-FRB construct and a 3xflag-FKBP-RILPL1 vector (Fig. 7 J). In the presence of rapamycin, FRB and FKBP dimerize, and RILPL1 translocates to the mitochondria outer membrane (Fig. 7, J and K). This leads to a significant clustering of the mitochondria toward the perinuclear area, reversed by pre-treatment with the dynein inhibitor dynarrestin (Fig. 7, K and L). Our data suggest that RILPL1 can act as a dynein adaptor protein through its interaction with dynactin, leading to clustering of LRRK2-positive lysosomes toward the centrosome and retraction of LYTL tubules.

### LYTL tubules elongate through tyrosinated microtubules

Motor proteins move organelles through microtubule tracks. We have previously shown that nocodazole disrupts JIP4-positive tubule formation, demonstrating that microtubules are important in tubule elongation. We confirmed the association between LYTL tubules and microtubules in live cells (Fig. S4 A and Video 3), showing a tubule elongating on a microtubule (Fig. S4 A, outlined arrowhead; open arrowhead on Video 3). -Tubulin undergoes critical posttranslational modifications (PTMs) that can regulate microtubule functions (for reviews, see [Nieuwenhuis and Brummelkamp, 2019; Roll-Mecak, 2020; Janke and Magiera, 2020]). Among them, acetylation on lysine 40 residue and tyrosination/detyrosination on its C-terminal tail are particularly important. Indeed, α-tubulin PTMs regulate the efficiency of motors, as motors have a preference for certain PTMs. For example, dynein has a preference

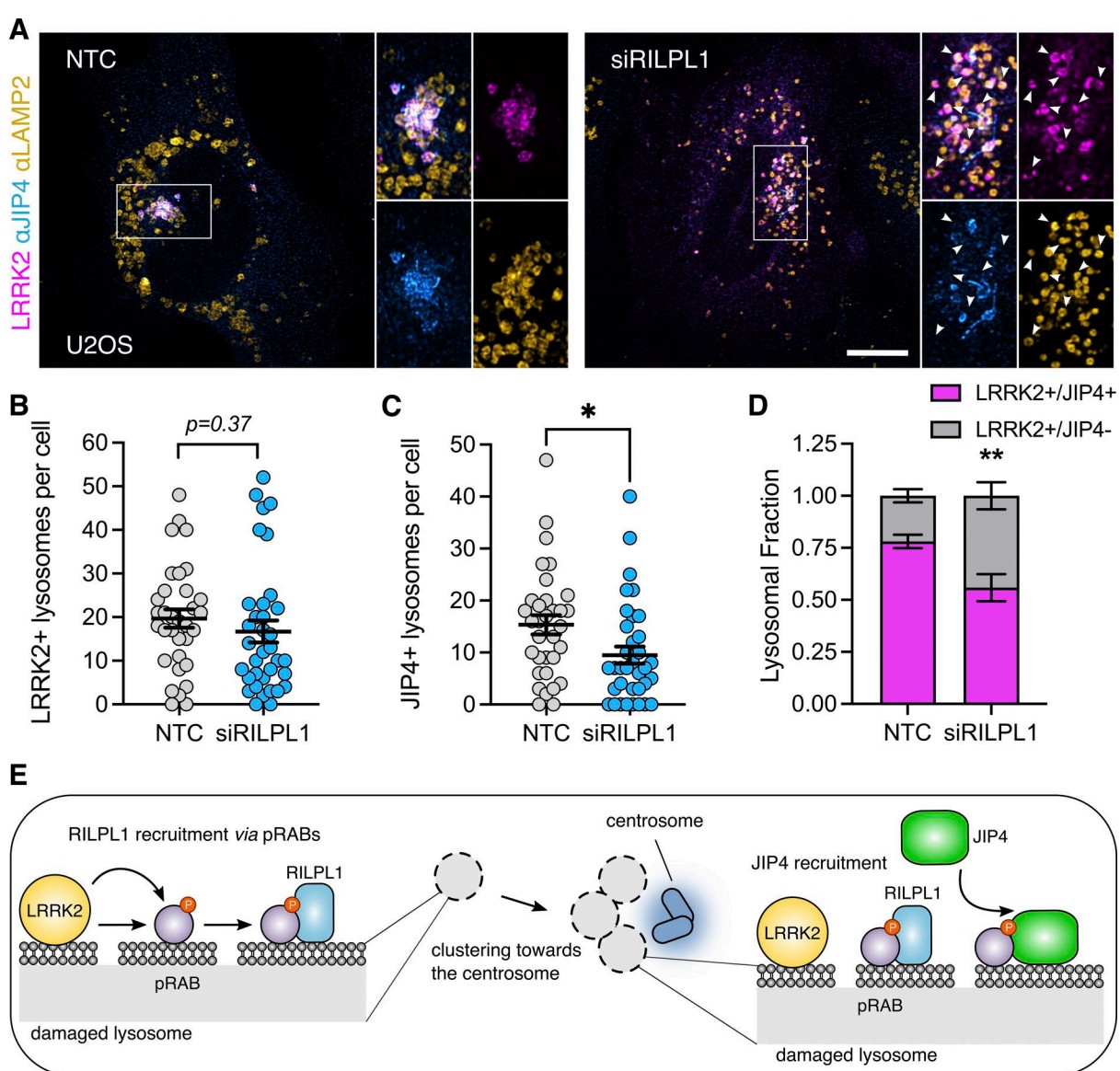

Figure 5. **RILPL1 depletion reduces JIP4 presence on LRRK2-positive lysosomes. (A)** U2OS cells were treated with NTC or siRILPL1 RNA. 24 h later, cells were transfected with HaloTag-LRRK2 for 36 h. Then, cells were treated with LLOME (2 h) and were fixed and stained with JIP4 and LAMP2 antibodies. **(B)** LRRK2+ lysosomes per cell were quantified. Data are mean ± SEM (n = 33–35 cells). Unpaired t test. **(C)** JIP4+ lysosomes per cell were quantified. Data are mean ± SEM (P = 0.0202, n = 33–35 cells). Unpaired t test. **(D)** The fraction of LRRK2+/JIP4+ lysosomes compared with LRRK2+/JIP4− lysosomes in NTC vs siRILPL1-treated cells. Data are mean ± SEM (P = 0.0035, n = 31–33 cells). Unpaired t test with Welch's correction for unequal variance. **(E)** Cartoon depicting the role of RILPL1 in clustering LRRK2+ lysosomes toward the centrosome and its effect on JIP4 recruitment. Arrowheads indicate LRRK2+/JIP4-lysosomes. Scale bar = 10 μm. NTC, nontargeting control.

for tyrosinated microtubules, and kinesin-1 family members prefer acetylated/detyrosinated microtubules. Acetylated α-tubulin is largely detyrosinated (Dunn et al., 2008; Katrukha et al., 2021) (Fig. S4 B), which allows the separation of most microtubules into two main groups: acetylated and tyrosinated (Tas et al., 2017; Guardia et al., 2016; Katrukha et al., 2021) (Fig. S4 B).

We therefore considered whether tubulin PTMs could affect tubule elongation. When analyzed in fixed cells, LYTL tubules show greater association with tyrosinated microtubules than acetylated microtubules (Fig. 8, A–C and Fig. S4 C), and live-cell imaging experiments show a JIP4-positive tubule growing on a tyrosinated microtubule (Fig. 8 D and Fig. S4 D). The sorted

material also travels through microtubules (Fig. S4 E and Video 4), and its movement is hampered by nocodazole (Fig. S4 F and Video 5). Time-lapse microscopy shows a fissioned tubule traveling on a tyrosinated microtubule (Fig. 8 E and Video 6). α-Tubulin tyrosination depends on tubulin tyrosine ligase (TTL) (Fig. 8 F) (Szyk et al., 2011), and it is known that TTL depletion leads to a dramatic increase in detyrosinated α-tubulin (Peris et al., 2006; Zink et al., 2012; Peris et al., 2009; Nieuwenhuis et al., 2017) (Fig. S5, A–D). TTL-depleted cells displayed lesser lysosomal tubulation than control cells (Fig. 8, G and H), consistent with the effect of nocodazole blocking tubulation (Bonet-Ponce et al., 2020). Nocodazole addition

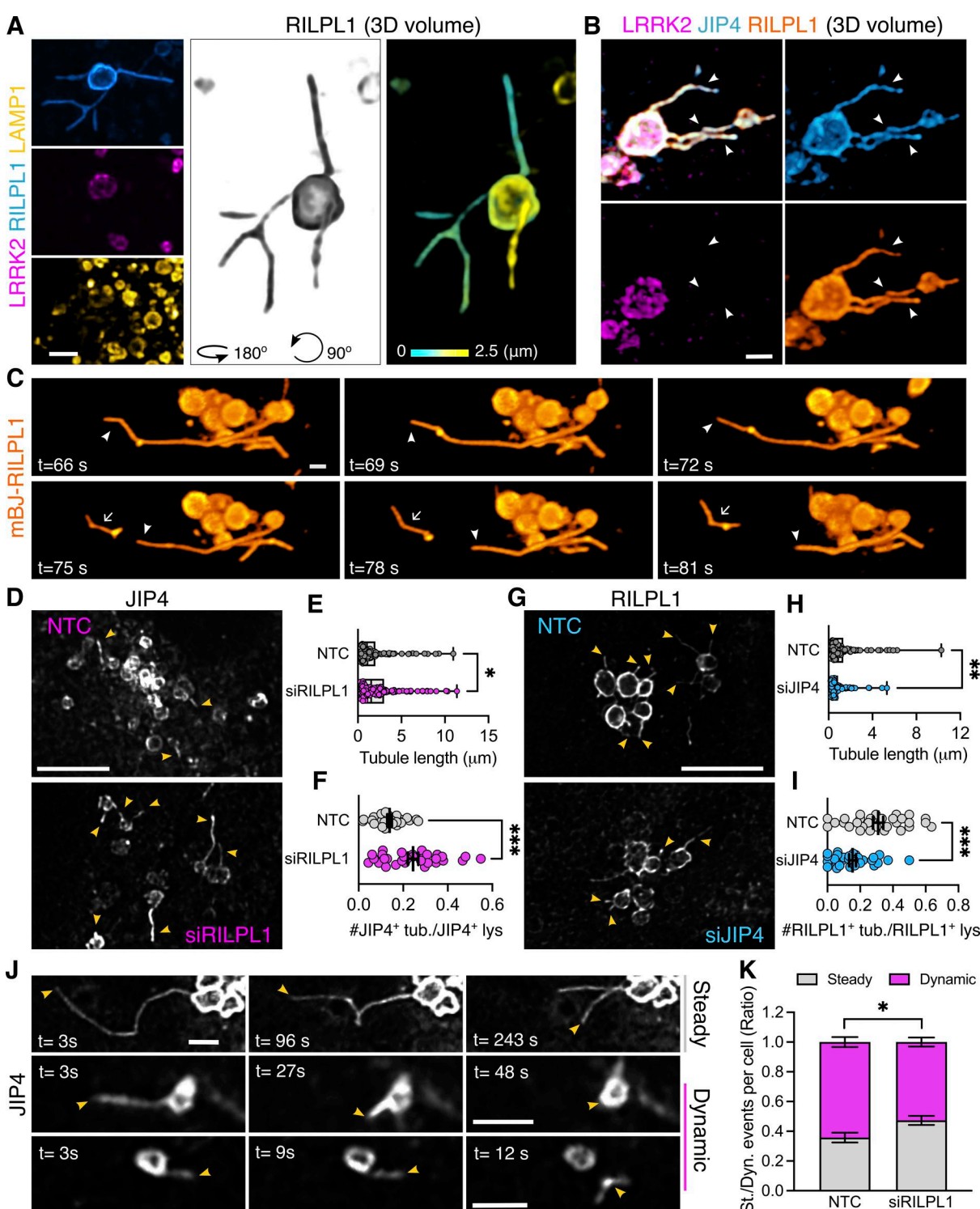

Figure 6. **RILPL1 colocalizes with LYTL tubules and reduces tubulation**. **(A)** Mouse primary astrocytes were transfected with HaloTag-LRRK2, mCherry-RILPL1, and LAMP1-mNeonGreen. 48 h later, cells were treated with LLOME for 6 h and imaged live. The 3D volume shows different RILPL1 tubules emanating from a lysosome. **(B)** U2OS cells were transfected with 3xflag-LRRK2, mBaoJin-RILPL1, and HaloTag-JIP4. 36 h later, cells were treated with LLOME for 2 h and fixed. 3D volume view of a lysosome with three LYTL tubules expressing JIP4 and RILPL1. **(C)** U2OS cells were transfected with 3xflag-LRRK2 and mBaoJin-RILPL1 for 36 h. Cells were then treated with LLOME and imaged live. Time-lapse shows RILPL1 staining dynamic tubules that undergo fission (3D view). **(D)** U2OS cells were treated with NTC or siRILPL1 for 24 h. Then cells were transfected with HaloTag-LRRK2 and mNeonGreen-JIP4 for 36 h. Cells were treated with LLOME for 2 h, fixed, and stained with an anti-LAMP1 antibody. **(E)** Box plot showing the tubule length in both groups. Unpaired t test was applied (P = 0.0458, n = 119–134 total tubules analyzed, respectively). Data are mean ± SEM. **(F)** Quantification of the number of JIP4+ tubules/total JIP4+ lysosomes. Data are mean ± SEM (P = 0.0002, n = 31–33 cells). Unpaired t test with Welch's correction was used. **(G)** U2OS cells were treated with NTC or siJIP4 for 24 h. Then cells were transfected with HaloTag-LRRK2 and 2xmyc-RILPL1 for 36 h. Cells were treated with LLOME for 2 h, fixed, and stained with anti-myc and anti-LAMP1 antibodies. **(H)** Box

plot showing the tubule length in both groups. Unpaired *t* test with Welch's correction was applied (P = 0.0037, *n* = 112–130 total tubules analyzed, respectively). Data are mean ± SEM. **(I)** Quantification of the number of RILPL1+ tubules/total RILPL1+ lysosomes. Data are mean ± SEM (P = 0.001, *n* = 31–32 cells). Unpaired *t* test with Welch's correction for unequal variance was used. **(J)** U2OS cells were treated with NTC or siRILPL1 for 24 h. Then cells were transfected with 3xflag-LRRK2 and mNeonGreen-JIP4 for 36 h. Cells were treated with LLOME for 2 h and analyzed live. Picture depicts examples of "steady" and "dynamic" tubular events. **(K)** Graph showing the ratio of steady and dynamic tubular events per cell in the different conditions. Data are mean ± SEM (P = 0.0141, *n* = 20 cells). Unpaired *t* test. Scale bar (D and G) = 10 µm; (A, B, and J) = 2 µm. NTC, nontargeting control.

depolymerizes the microtubule network, leaving few nocodazole-resistant microtubules (Xu et al., 2017) (Fig. S5 E), which are acetylated/detyrosinated and not tyrosinated (Xu et al., 2017; Kesarwani et al., 2020) (Fig. S5 F). Interestingly, RILPL1 knockdown significantly decreases α-tubulin tyrosination levels (Fig. 8, I and J).

Taken together, our data show that LYTL tubules move through tyrosinated microtubules, which is consistent with the observation that the dynein/dynactin complex requires α-tubulin tyrosination to initiate their retrograde movement (McKenney et al., 2016; Nirschl et al., 2016). As LYTL tubule retraction is mediated by the RILPL1:dynein/dynactin complex, tubule elongation to the plus-end of microtubules is likely driven by JIP4 and kinesin(s). Even though JIP4 has long been associated with KIF5B (a kinesin-1 family member), KIF5B knockdown has no effect on tubulation (Fig. S5, G and H). In addition, we could not detect an effect in tubulation upon KIF1A/B knockdown, two other kinesins previously associated with lysosomes (Guardia et al., 2016), suggesting that another kinesin is responsible for tubule elongation. Our data are consistent with the fact that kinesin-1 family members preferentially move on acetylated/detyrosinated microtubules. It is therefore likely that the kinesin(s) associated with JIP4 in LYTL tubule elongation have preference for tyrosinated microtubules (Fig. 9).

## Discussion

Lysosomes change the shape of their membrane to adapt to various cellular responses and undergo distinct mechanisms of membrane tubulation. For example, in the context of immune cell response to LPS treatment, lysosomes undergo a transformation from vesicular to tubular morphology, a change essential for MHC-II presentation (Chow et al., 2002; Saric et al., 2016). When the lysosomal pool becomes saturated, lysosomes generate new lysosomes through a process known as lysosomal reformation (LR) (Yu et al., 2010). In LR, new lysosomes emerge through the extrusion of membranes from preexisting healthy lysosomes, featuring membrane markers such as LAMP1, and this process is facilitated by the motor protein KIF5B (Du et al., 2016). We have previously described LYTL as a novel lysosomal tubulation and vesicle sorting process that is orchestrated by the Parkinson's disease kinase LRRK2. We hypothesize that LYTL originates from compromised lysosomes and could serve as a mechanism for transferring undegraded material from dysfunctional lysosomes to active ones. However, we are still working to identify the cargo carried by LYTL vesicles.

In this study, we systematically mapped the LRRK2 lysosomal proteome. Notably, our screening highlighted RAB proteins as the most abundant group, confirming prior data that these proteins are the primary targets of active LRRK2 on lysosomes and underscoring the reliability of our screening approach. Among the proteins identified, two potential pRAB effectors, JIP4 and RILPL1, were observed. Given their roles as motor adaptor proteins in LYTL, it is reasonable to consider that LYTL is the major consequence of the accumulation of LRRK2 on dysfunctional lysosomes. It is also worth noting that calcium-binding proteins and RNA-binding proteins were also hits in this screen and may be of interest to follow up in additional studies.

RILPL1 has been previously linked to LRRK2 in other cellular compartments, including the centrosome and the primary cilia (Lara Ordónez et al., 2019; Dhekne et al., 2018). It has been proposed that hyperactive LRRK2 disrupts centrosomal cohesion and prevents ciliogenesis via pRABs and RILPL1. In agreement with our results, recent work also demonstrates an LRRK2-dependent recruitment of RILPL1 to lysosomes in cells harboring the VPS35-D620N mutation (Pal et al., 2023). However, the molecular function of RILPL1 has not been clarified. Here, we show that RILPL1 clusters LRRK2-positive lysosomes around the centrosome and reduces LYTL-dependent tubulation. RILPL1 binds to p150[Glued], the largest subunit of dynactin, which binds to both microtubules (Waterman-Storer et al., 1995) and cytoplasmic dynein (Vaughan and Vallee, 1995), promoting retrograde transport of cargo to the minus-end of microtubules (Waterman-Storer et al., 1997). Our data demonstrate that RILPL1 promotes lysosomal retrograde transport and tubule retraction via the dynein/dynactin complex. The possibility that RILPL1 might act as a motor adaptor protein aligns with the observed role of the other members of the RHD family, all of which are known to share this function. Our data suggest a very tight regulation of LYTL dynamics, with both motor adaptor proteins working in opposite directions. It is worth noting that a competition between both effector proteins is unlikely, as JIP4 overexpression increases the RILPL1:pRAB10 interaction, suggesting a further stabilization of the complex. Although a direct interaction between RILPL1 and JIP4 was not detected, the relationship between JIP4 and RILPL1 is worth exploring in future endeavors. It is also notable the need to find other LYTL tubule markers independent of JIP4 and RILPL1 to be able to identify all possible tubulation events.

The dynein/dynactin complex relies on tubulin tyrosination to initiate retrograde movement (McKenney et al., 2016; Nirschl et al., 2016), and, as shown here for the first time, LYTL tubules extend along tyrosinated microtubules. Since JIP4 is essential for promoting the elongation of LYTL tubules toward the plus-end of microtubules, it is plausible that JIP4 interacts with kinesin proteins to facilitate this process. When JIP4 is recruited to membranous compartments by ARF6, JIP4 preferentially binds to dynein/dynactin (Montagnac et al., 2009; Cason and Holzbaur,

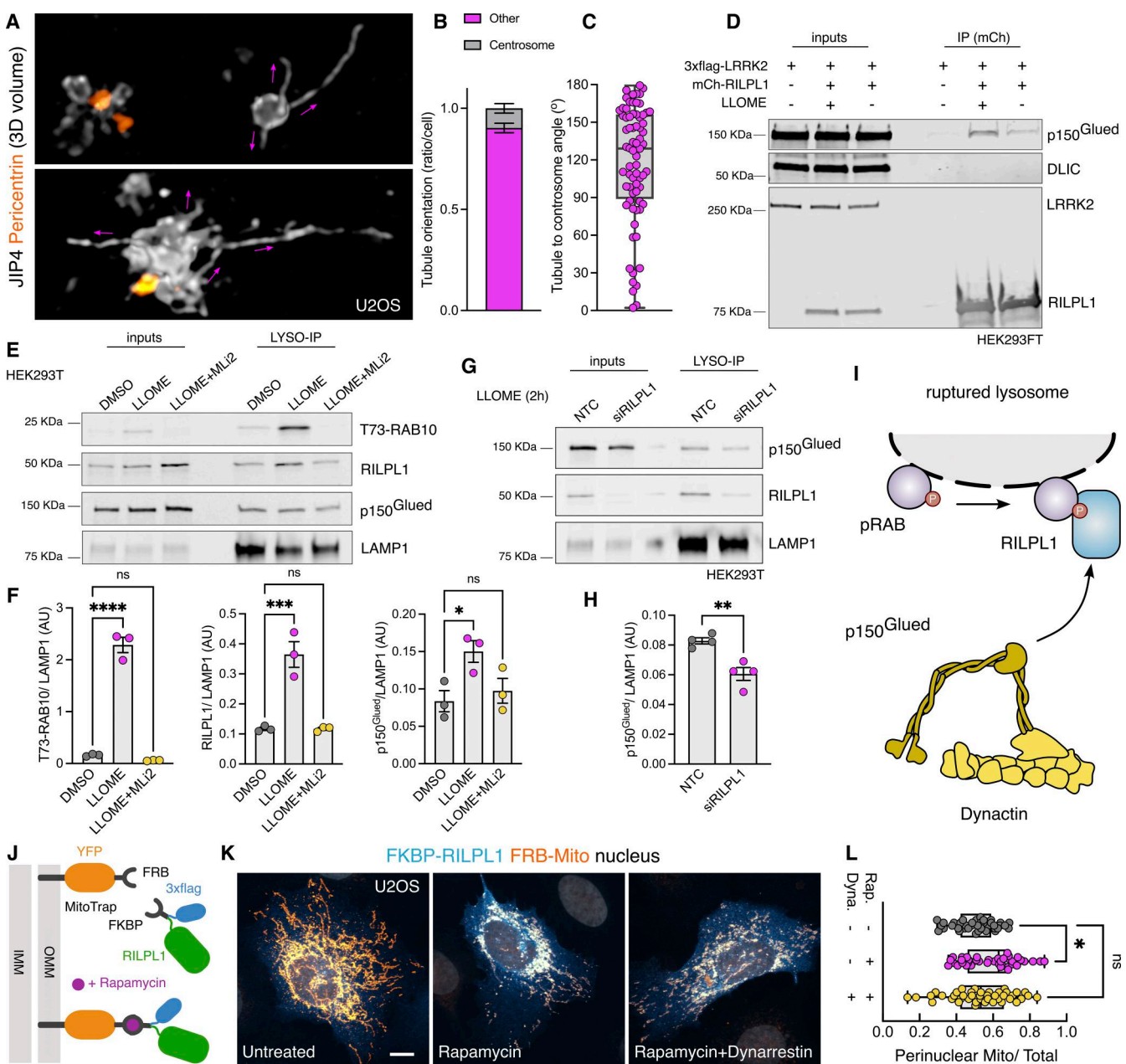

**Figure 7. RILPL1 binds to p150<sup>Glued</sup> and favors its recruitment to lysosomes. (A)** U2OS cells were transfected with HaloTag-LRRK2 and mNeonGreen-JIP4 for 36 h. Cells were then treated with LLOME (2 h), fixed, and stained with pericentrin. Volume view image of JIP4+ tubules and centrosomes (pericentrin). **(B)** The orientation of the JIP4+ tubules was manually annotated as "toward the centrosome" (centrosome) or "elsewhere" (other) (n = 28 tubules). **(C)** Box plot depicting the tubule-to-centrosome angle (n = 72 tubules from 14 cells). **(D)** HEK293FT cells were transfected with 3xflag-LRRK2 and mCherry-RILPL1 for 24 h. Cells were treated or not with LLOME for 2 h, and lysates were subjected to immunoprecipitation with anti-RFP antibodies. **(E)** HEK293T cells stably expressing GFP-LRRK2 and TMEM192-3xHA were seeded and treated with DMSO, LLOME, or LLOME + MLi2 24 h later. Lysosomes were purified with anti-HA beads following the LYSO-IP technique. Lysosomes were then lysed, and their content was analyzed via immunoblotting. **(F)** Quantification of T73-RAB10, RILPL1, and p150<sup>Glued</sup> protein levels from the lysosomal fraction. Data are mean ± SEM from three independent replicates. P (T73-RAB10) <0.0001; P (RILPL1) = 0.0008; P (p150<sup>Glued</sup>) = 0.0368. One-way ANOVA with Dunnett's. **(G)** HEK293T cells stably expressing GFP-LRRK2 and TMEM192-3xHA were seeded and treated with NTC and siRILPL1 for 48 h later. Then, cells were treated with LLOME for 2 h. Lysosomes were purified with anti-HA beads following the LYSO-IP technique. Lysosomes were then lysed, and their content was analyzed via immunoblotting. **(H)** Quantification of p150<sup>Glued</sup> protein levels from the lysosomal fraction. Data are mean ± SEM from four independent replicates. P = 0.0036. Unpaired *t* test. **(I)** Schematic representation of our experimental model where the pRAB proteins recruit RILPL1, which in turn binds to p150<sup>Glued</sup> and recruits it to membrane-damaged lysosomes. **(J)** Cartoon explaining the FRB–FKBP system to trap RILPL1 to the outer mitochondrial membrane (OMM). **(K)** U2OS cells were transfected with Mito-YFP-FRB and 3xflag-FKBP-RILPL1 for 24 h. Cells were treated or not with rapamycin or rapamycin+dynarrestin. **(L)** Box plot showing the perinuclear Mito/Total ratio in all groups. Data are mean ± SEM (P = 0.0198, n = 39–42 cells). Brown–Forsythe and Welch ANOVA with Dunnett's post hoc test. Scale bar = 10 μm. NTC, nontargeting control. Source data are available for this figure: SourceData F7.

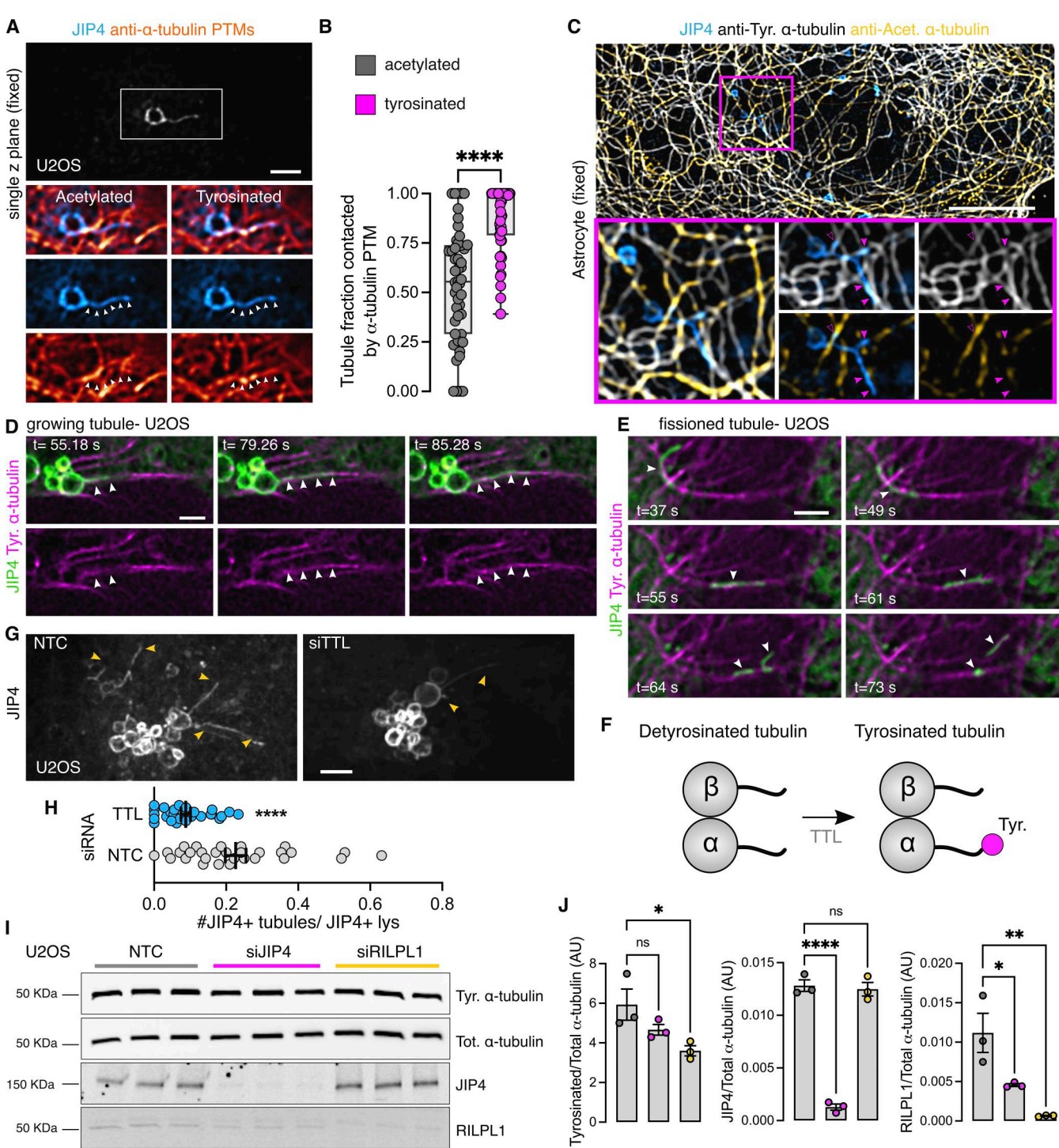

Figure 8. **LYTL tubules elongate through tyrosinated microtubules. (A)** U2OS cells were transfected with 3xflag-LRRK2 and GFP-JIP4. Cells were treated with LLOME (2 h) and fixed. After fixation, cells were stained for GFP, tyrosinated α-tubulin, and acetylated α-tubulin. **(B)** Box plot showing the LYTL tubule contact fraction with the different α-tubulin PTMs. Unpaired t test with Welch's correction was applied (P < 0.0001, n = 49 tubules). Arrowheads show lysosomal tubules. Scale bar = 2 μm. **(C)** Mouse primary astrocytes were transfected with 3xflag-LRRK2 and GFP-JIP4. Cells were treated with LLOME (6 h) and fixed. After fixation, cells were stained for GFP, tyrosinated α-tubulin, and acetylated α-tubulin. **(D and E)** Time-lapse experiment of U2OS cells transfected with 3xflag-LRRK2, mNeonGreen-JIP4, and TagRFP-T-A1aY1 (tyrosinated microtubules) and treated with LLOME for 2 h before imaging. **(F)** The enzyme TTL is responsible for adding the tyrosine residue into the C-terminal tail of α-tubulin. **(G)** U2OS cells were incubated with a nontargeting control (NTC) or TTL siRNA and transfected 24 h later with 3xflag-LRRK2 and mNeonGreen-JIP4. Cells were treated with LLOME (2 h) and imaged live. The #JIP4+ tubules/JIP4+lysosomes ratio was quantified. **(H)** Graph depicts the ratio in the NTC and siTTL groups. Unpaired t test with Welch's correction was applied. Data are mean ± SEM (P < 0.0001, n = 27–29 cells). Magenta arrowheads show colocalization between LYTL tubules and only tyrosinated microtubules (filled) and tyrosinated + acetylated microtubules (empty). White arrowheads in A show the length of a LYTL tubule; in D, show a budding tubule elongating on a tyrosinated microtubule; and in E, a fissioned tubule traveling to a tyrosinated microtubule. Yellow arrowheads indicate JIP4-positive tubules. **(I)** U2OS cells were treated with NTC,

JCB

siJIP4, and siRILPL1 siRNA for 72 h and lysed. Lysates were blotted against anti-tyrosinated α-tubulin, total α-tubulin, JIP4, and RILPL1 antibodies. **(J)** Histograms showing tyrosinated α-tubulin, JIP4, and RILPL1 protein levels, normalized to total α-tubulin. Data are mean ± SEM from three independent replicates. One-way ANOVA with Dunnett's post hoc test. Scale bar (C) = 10 μm; (A and D–G) = 2 μm. Insets = 2 μm. Source data are available for this figure: SourceData F8.

2023). However, our data suggest that once JIP4 is recruited to the lysosomal surface by pRABs, JIP4 binds to kinesin. This observation aligns with the other studies of the role of pRABs and JIP4 in neuronal autophagosomal transport (Boecker et al., 2021; Dou et al., 2023; Cason and Holzbaur, 2023). Although JIP4 has primarily been associated with members of the kinesin-1 family, knockdown of KIF5B did not result in a decrease in LYTL tubulation. This finding suggests that KIF1A, KIF1B, and KIF5B do not play a role in tubule elongation and that JIP4 binds to a different kinesin in this model, consistent with the fact that kinesin-1 family members travel on acetylated/detyrosinated microtubule tracks. The kinesin(s) responsible for tubule elongation are therefore still unknown and will be addressed in future work. The lack of RILPL1 leads to a decrease in tubule retraction and fission. Given the differential recruitment of motors by RILPL1 and JIP4, respectively, the combined retraction and elongation force might create a more favorable environment for tubule fission into moving vesicles.

In summary, here we have explored the role of RILPL1 once recruited by LRRK2 to dysfunctional lysosomes. We describe new regulatory proteins of LYTL, which is likely pathologically relevant given the role of LRRK2 and lysosomes in PD.

## Materials and methods
### Cell culture
U2OS cells (ATTC), HEK293FT, HEK293T, and mouse primary astrocytes were maintained in DMEM (Thermo Fisher Scientific) containing 4.5 g/Liter glucose, 2 mM l-glutamine, and 10% FBS (Gibco) at 37°C in 5% $CO_2$ into 75-$cm^2$ tissue culture flasks. All cell lines were used until passage 30. Cells were seeded on 12-mm coverslips for fixed experiments and in 35-mm glass bottom dishes (Mattek) for live-cell imaging experiments. Coverslips and dishes were pre-coated with Matrigel (Corning).

Primary astrocyte cultures were prepared from C57BL/6J newborn (postnatal day 0) pups. Dissected mouse cortices were incubated in 1 ml/cortex of Basal Medium Eagle (BME) (Sigma-Aldrich), containing 5 U of papain (Worthington) for 30 min at 37°C. Five micrograms of deoxyribonuclease I were added to each cortex preparation, and brain tissue was dissociated into a cellular suspension that was washed twice with 10 volumes of BME and counted. Astrocyte cultures were plated in DMEM (Thermo Fisher Scientific), supplemented with 10% FBS (Gibco), into 75-$cm^2$ tissue culture flasks. For the preparation of purified

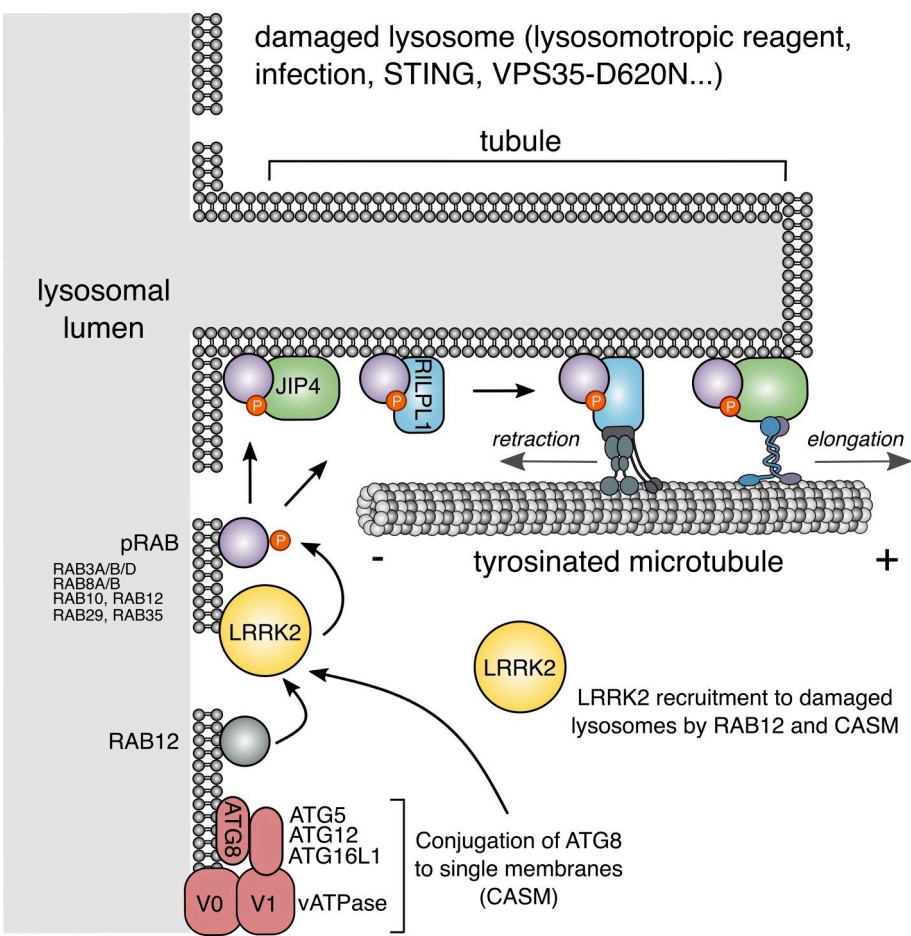

Figure 9. **Schematic representation of our working model.** LRRK2 is recruited to ruptured lysosomes (likely via RAB12 and CASM), where it phosphorylates and recruits RAB substrates. pRABs recruit their effectors JIP4 and RILPL1. JIP4, through a yet unknown kinesin, elongates the LYTL tubules, whereas RILPL1, via dynein/dynactin, favors tubule retraction.

astrocyte cultures, 7- to 10-day primary cultures were vigorously shaken to detach microglia and oligodendrocytes.

The HEK293T-inducible GFP-LRRK2-WT cell line was obtained from D. Alessi (University of Dundee, Dundee, UK), and expression was induced by addition of doxycycline (Nichols et al., 2010).

## Reagents and treatments

LLOME (L7393; Sigma-Aldrich; and HY-129905A; MedChemExpress) was diluted in DMSO and added at 1 mM for the indicated times. MLi-2 (5756; Tocris; and HY-100411; MedChemExpress) was used at 1 µM in DMSO for 90 min prior to LLOME incubation. Nocodazole (M1404; Sigma-Aldrich) was diluted in DMSO and added at 10 µM, 30 min before imaging. HaloTag–transfected cells were incubated with the JFX650 peptide (Janelia) at 100 nM for 10 or 40 min, and cells were then washed three times and incubated with fresh medium before being treated with LLOME. For experiments utilizing the FKBP–FRB complex, rapamycin (3346; Cayman Chemicals) was added at 200 nM for 2 h prior to fixation in 4% PFA. Dynarrestin (6526; Tocris) was added at 25 µM, 30 min before rapamycin.

## Antibodies

The following primary antibodies were used: mouse anti-FLAG M2 (F3165; Sigma-Aldrich; 1:500 for ICC and 1:10,000 for WB), rabbit anti-JIP4 (5519, 1:1,000; Cell Signaling Technology for WB and 1:100 for ICC), rabbit anti-LAMP1 (D2D11; 1:2,000; Cell Signaling Technologies for WB and 1:200 for ICC), rat anti-LAMP1 (Developmental Studies Hybridoma Bank [DSHB], 1D4B; 1:100 for ICC), mouse anti-LAMP1 (H4A3; 1:1,000; DSHB for WB and 1:100 for ICC), mouse anti-LAMP2 (H4B4; 1:1,000; DSHB for WB and 1:100 for ICC), mouse anti-p150$^{Glued}$ (610474; 1:10,000; BD Biosciences for WB), rabbit anti-DLIC (A304-208A; 1:1,000; Bethyl Laboratories for WB), chicken anti-GFP (GFP-1020; 1:500 to 1:1,000; Aves Lab for ICC), mouse anti-GFP (11814460001; 1:10,000; Roche for WB), mouse anti–α-tubulin (3873; 1:10,000; Cell Signaling Technology for WB), rat anti-tyrosinated α-tubulin (YL1/2) (ab6160; 1:10,000; Abcam for WB and 1:500 for ICC), rabbit anti-detyrosinated α-tubulin (ab48389; 1:1,000; Abcam for WB and 1:100 for ICC), mouse anti-acetylated α-tubulin (T7451: 1:200; Millipore-Sigma for ICC), rabbit anti-total α-tubulin (ab52866; 1:10,000; Abcam for WB), rabbit anti-RILPL1 (MJF-R41-21 [ab302492]; 1:100; Abcam for ICC and 1:1,000 for WB), rabbit anti-RAB10 (ab237703; 1:100; Abcam for ICC and 1:2,000 for WB), rabbit anti-RAB10 (phospho-T73) (MJF-R21-22-5 [ab241060]; 1:100; Abcam for ICC and 1:2,000 for WB), rabbit anti-RAB12 (phosphor-S106) (MJF-R25-9 [ab256487]); 1:1,000; Abcam for WB), rabbit anti-LRRK2 (ab133474; 1:1,000; Abcam for WB), rat anti-HA (7C9) (7c9-100; 1:10,000; ChromoTek for WB and 1:500 for ICC), rat anti-myc (9E1; 1:20,000; ChromoTek for WB and 1:500 for ICC), rabbit anti-EEA1 (3288; 1:100; Cell Signaling Technology for ICC), rabbit anti-LC3B (2775; 1:1,000; Cell Signaling Technology for WB), rabbit anti-LAMTOR4 (12284; 1:1,000; Cell Signaling Technologies for WB), rabbit anti-calreticulin (D3E6) (12238; 1:1,000; Cell Signaling Technologies for WB), mouse anti-GM130 (H7) (sc-55590; 1:1,000; Santa Cruz Biotechnology for WB),

rabbit anti-GAPDH (G9545; 1:10,000; Millipore-Sigma for WB), rabbit anti-Cathepsin D (219361, 1:2,000; Millipore for WB), mouse anti-pericentrin (ab28144; 1:100; Abcam for ICC), and rabbit anti-Tom20 (FL-145) (sc-11415; 1:5,000; Santa Cruz Biotechnology for WB).

For ICC, unless otherwise stated, the secondary antibodies were purchased from Thermo Fisher Scientific. The following secondary antibodies were used: donkey anti-mouse Alexa Fluor 568 (A10037, 1:500), donkey anti-rabbit Alexa Fluor 488 (A-21206, 1:500), donkey anti-mouse Alexa Fluor 568 (A-21202, 1:500), donkey anti-rat Alexa Fluor 488 (A-21208, 1:500), donkey anti-rabbit Alexa Fluor 568 (A10042, 1:500), donkey anti-mouse Alexa Fluor 647 (A-31571, 1:500), and goat anti-rat Alexa Fluor 647 (A-21247, 1:250 to 1:500). Donkey anti-chicken Alexa Fluor 488 (703-545-155, 1:500) and donkey anti-rat Alexa Fluor 405 (712-475-153, 1:100) were obtained from Jackson ImmunoResearch.

## Plasmids

Constructs for 3xFLAG-LRRK2, HaloTag-LRRK2, LYSO-LRRK2, mNeonGreen-JIP4, pDEST-53-JIP4, HaloTag-JIP4, 2xmyc-RAB10 (WT and T73A), and LAMP1-HaloTag have been described previously (Beilina et al., 2014; Beilina et al., 2020; Bonet-Ponce et al., 2020; Bonet-Ponce and Cookson, 2022b; Kluss et al., 2022a). RILPL1 cDNA was amplified with PCR and cloned into the pCR8/GW/TOPO vector (Thermo Fisher Scientific). PCR8-RILPL1 was then subcloned into pDEST-3xflag-FKBP, pDEST-mBaoJin, and pDEST-2xmyc using Gateway Technology (Thermo Fisher Scientific). mCherry-RILPL1 was created with IN-FUSION (Takara) from the mCherry-Climp63 vector (#136293; Addgene) (Shibata et al., 2008). RILPL1-R293A was created using the QuickChange Lightning Site-Directed Mutagenesis Kit (Agilent) from the mCherry-RILPL1 (WT) vector. Ubc-3xHA-JIP4 was created by adding the Ubc promoter to a pDEST-pGCS-3xHA plasmid and then performing an LR reaction (Gateway Technology) to subclone JIP4 *via* PCR8-JIP4.

LAMP1-RFP, LAMP1-mNeonGreen, TMEM192-3xHA, LifeAct-mNeonGreen, 6xHis-TagRFP-T_A1aY1, pEYFP-Mitotrap, TUBB5-Halo, and EMTB-mNeonGreen plasmids were purchased from Addgene (#1817; Addgene, #98882; Addgene, #102930; Addgene, #98877; Addgene, #158754; Addgene, #46942; Addgene, #6469; Addgene, #137802; Addgene) (Sherer et al., 2003; Chertkova et al., 2017, *Preprint*; Abu-Remaileh et al., 2017; Kesarwani et al., 2020; Robinson et al., 2010; Uno et al., 2014).

## Transfection

Transient transfections of U2OS and mouse primary astrocyte cells were performed using Lipofectamine Stem Reagent and Opti-MEM (Thermo Fisher Scientific). HEK293T and HEK293FT cells were transfected using Lipofectamine 2000 and Opti-MEM (Thermo Fisher Scientific). HEK293T and HEK293FT cells were transfected for 24 h, and U2OS cells were transfected for 36–48 h prior to imaging. Astrocytes were transfected for 48 h prior to imaging. For siRNAs, cells were transfected with the SMARTpool ON-TARGET (Horizon) plus scramble, or TTL, JIP4, RILPL1, KIF1A, KIF1B, and KIF5B siRNA using Lipofectamine RNAiMAX (Thermo Fisher Scientific) transfection

reagent. U2OS cells were incubated with siRNA for a total of 3 days before imaging or lysis. HEK293T cells were incubated with siRNA for 48 h prior to performing the LYSO-IP.

## GFP-LRRK2/TMEM193-3xHA stable cell production
HEK293T cells stably expressing GFP-LRRK2 were transfected with TMEM192-3xHA using Lipofectamine 2000 at a 75% confluency in a 75-cm$^2$ flask. 24 h later, cells were treated with puromycin (1 µg/ml) for 7 days. The polyclonal population was seeded in a density of one cell per well in 96-well plates. The presence of TMEM192-3xHA was corroborated by staining and immunoblotting in the selected clone.

## LYSO-IP
Lysosomes from HEK293T cells stably expressing GFP-LRRK2 and TMEM192-3xHA were purified following the protocol described by Davis et al. (2021). Briefly, cells were seeded in 15-cm dishes (proteomics) or 10-cm dishes (immunoblotting) at a density appropriate for them to reach confluency after 24 or 48 h (for siRNA experiments). All subsequent steps were performed on ice or at 4°C unless otherwise noted. After media removal, cell monolayers were rinsed with ice-cold KPBS buffer (136 mM KCl, 10 mM KH2PO4, pH 7.25, supplemented with 1x protease and phosphatase inhibitor cocktail [Thermo Fisher Scientific]), scraped into 10 ml (proteomics) or 1 ml (immunoblotting) of KPBS, and collected by centrifugation at 300 $g$ for 5 min. Pelleted cells were resuspended in a total volume of 1 ml KPBS (supplemented with 3.6% [wt/vol] Opti-Prep [Sigma-Aldrich]) and fractionated by passing through a 23-G syringe five times followed by centrifugation at 700 $g$ for 10 min. Postnuclear supernatant was harvested and incubated with 100 µl (proteomics) or 40 µl (immunoblotting) of anti-HA magnetic beads (Thermo Fisher Scientific, prewashed with KPBS buffer) with end-over-end rotation for 15 min at 4°C. Lysosome-bound beads were washed two times with KPBS(+OptiPrep) and two times with KPBS. Samples were incubated with lysis buffer (20 mM Tris-HCl [pH 7.5], 150 mM NaCl, 1 mM EDTA, 0.3% Triton X-100, 10% glycerol, and 1× Halt protease and phosphatase inhibitor cocktail) for 25 min with end-over-end rotation at 4°C. Beads were removed, and lysates were snap-frozen with LN2 (proteomics) or boiled at 95°C for 5 min with 4x loading buffer (Bio-Rad) and β-mercaptoethanol (immunoblotting).

## Mass spectrometry analysis
In-gel samples were reduced with 5 mM Tris(2-carbox-yethyl)phosphine (Sigma-Aldrich) and alkylated with 5 mM N-Ethylmaleimide (NEM), (Sigma-Aldrich). Samples were digested with trypsin (Promega) at a 1:10 (wt/wt) trypsin:sample ratio at 37°C for 18 h. Peptides were extracted and then desalted using an Oasis HLB plate (Waters). An Orbitrap Lumos mass spectrometer (Thermo Fisher Scientific) coupled in line with an Ultimate 3000 HPLC (Thermo Fisher Scientific) was used for data acquisition. Peptides were directly injected and separated on an ES803A column (75-µm inner diameter, 25-cm length, 2-µm C18 beads; Thermo Fisher Scientific) using a gradient with mobile phase B (0.1% formic acid in LC-MS grade acetonitrile)

increased from 5 to 25% in 90 min at a flow rate of 300 nl/min. LC-MS/MS data were acquired in data-dependent mode. The survey scan was performed at a resolution of 120k and a mass range of 400–1,500 m/z. MS2 data were acquired in an ion trap with a rapid scan rate and an isolation window of 1.6 m/z. The CID method with a fixed collision energy of 30 was used for peptide fragmentation. The minimum signal intensity required to trigger the MS2 scan was 1e4.

Raw data were processed using Proteome Discoverer 2.4 Software. Data were searched against the UniProt Human database. The mass tolerances for precursor and fragment were set to 5 ppm and 0.6 Da, respectively. Up to 1 missed cleavage was allowed. NEM on cysteines was set as a fixed modification. Variable modifications include Oxidation (M), Met-loss (Protein N-term), and Acetyl (Protein N-term). 1% false discovery rate at the protein level was applied. Proteins detected with 1 peptide were further filtered out from the results. The protein abundances were obtained by summing the abundance of the connected peptides, then normalized to the total peptide amount. Protein ratios were calculated by comparing the median protein abundance of each group. The ANOVA (Individual Proteins) method was used for the hypothesis test.

## Confocal microscopy
### Airyscan
Airyscan images were taken using a Zeiss LSM 880 microscope equipped with a 63× 1.4 NA objective. Super-resolution imaging was performed using the Airyscan mode. Raw data were processed using Airyscan processing in "auto strength" mode with Zen Black software version 2.3.

### SoRa
For spinning disk super-resolution microscopy, we used a W1-SoRa super-resolution spinning disk microscope (Nikon) with a 60× 1.49 NA oil immersion objective. A 2.8× intermediate magnification (168X combined) was used in time-lapse experiments regarding JIP4 tubules. A 4× intermediate magnification (240X combined) was used in snapshot images. An offset micro-lensed SoRa disk was used, and an environmental chamber maintained cells at 37°C with humidified 5% $CO_2$ gas during imaging. For deconvolution, we used 20–25 iterations of the 3D Landweber or 3D blind algorithms with the NIS-Elements AR 5.21.03 software. Images from two channels were acquired simultaneously using a Cairn twin-cam emission splitter and two Photometrics prime 95b sCMOS cameras, a 565LP DM, and appropriate emission cleanup filters. Triggered piezo was used to maximize speed. Stacks were taken with 0.2-µm distance between slices.

When needed, bleaching was corrected using the "Histogram matching" option from Fiji (ImageJ). Unless otherwise stated, stacks were processed as maximum intensity projection.

## Lookup tables
All pseudocolors used in this paper can be found in the "NeuroCyto LUTs" collection (Fiji, ImageJ). For gray colors, "JDM grays" were used. For cyan colors, "cyan," "cyan hot," and "JDM duo cyan" were used. For magenta, "Magenta" and "Magenta Hot" were used. For orange, "NanoJ-Orange" was used. For

yellow, "Yellow Hot" and "JDM duo intense yellow" were used. For inverted Lookup tables (LUTs), "JDM gamma inverted" was used. For the z color code, "Z-stack Depth Colorcode 0.0.2" plugin (Fiji, ImageJ) was applied using "DavLUT-Bright" as LUT.

## Volume view
Volume view was obtained using the NIS-Elements AR 5.21.03 software (Nikon) with the "Alpha Blending" option with or without "Depth Color Code." Volume view was also obtained with the "volume viewer" plugin from Fiji (ImageJ; NIH).

# (JIP4 or RILPL1$^+$) tubules/(JIP4 or RILPL1$^+$) lysosomes measurements.

U2OS cells were transfected with the 3xflag-LRRK2 plasmid and co-transfected with mNeonGreen-JIP4. After treatment with LLOME, live cells were imaged with SoRa. The tubular ratio in each cell was measured as: Ratio = #JIP4+tubules/#JIP4+lysosomes. Only cells with 10 or more JIP4-positive lysosomes per cell were imaged.

## LYTL tubule contact to α-tubulin PTM
U2OS cells were transfected with 3xflag-LRRK2 and pDEST53-JIP4 for 36 h. After treatment with LLOME for 2 h, cells were fixed and stained for GFP, tyrosinated α-tubulin, and acetylated α-tubulin. "Tubule fraction contacted by α-tubulin PTM" was measured as: distance of contact between LYTL tubule and α-tubulin PTM/total distance of LYTL tubule.

## Perinuclear (lysosomes or mitochondria)/total (lysosomes or mitochondria)
LAMP1 and Mito-YFP integrated densities were measured using Fiji (ImageJ; NIH) after thresholding. Every cell was thresholded according to its individual intensity, capturing the whole lysosomal or mitochondrial population. Integrated density was measured for the whole cell (Total), and the area within 5 μm of the nucleus (Perinuclear). "Perinuclear/Total" ratio was measured as: PR = Perinuclear/Total.

## Tubule-to-centrosome angle
To ascertain if LYTL tubules are oriented toward the centrosome, the tubule-to-centrosome angle was measured. Briefly, a segmented line was traced following LYTL tubules from their tip to their base and then from the base to the centrosome (stained with pericentrin). The angle of the segmented line was measured with Fiji (ImageJ; NIH). A tubule oriented toward the centrosome would have an angle close to 0°.

## Steady/dynamic tubular event measurement
U2OS treated with NTC or siRILPL1 RNA for 24 h were transfected with 3xflag-LRRK2 and mNeonGreen-JIP4 for 36 h. Cells were then treated with LLOME for 2 h and imaged live. Stacks were taken every 3 s for a full duration of 4 min. LYTL tubular events were categorized as follows: Steady (when tubules did not undergo significant change in the full duration of the movie) and dynamic (when tubules retracted completely until their base, or underwent sorting). Steady events recorded for <30 s from elongation to the end of the movie were not taken into consideration. Only cells with five or more JIP4-positive tubules were imaged.

## Immunostaining
### Imaging fixed LYTL tubules in fixed cells
Cells were fixed with 4% PFA for 10 min, permeabilized with PBS/0.1% Triton for 10 min, and blocked with 5% donkey serum for 1 h at RT. Primary antibodies were diluted in blocking buffer (1% donkey serum) and incubated overnight at 4°C. After three 5 min washes with PBS/0.1% Triton, secondary fluorescently labeled antibodies were diluted in blocking buffer (1% donkey serum) and incubated for 1 h at RT. Coverslips were washed twice with 1x PBS and an additional two times with dH$_2$O and mounted with ProLong Gold antifade reagent (Thermo Fisher Scientific).

### Imaging microtubules in fixed cells
Cells were fixed in methanol for 5 min at –20°C and washed with PBS three times. Cells were permeabilized with PBS/0.1% Triton for 10 min and blocked with 5% donkey serum for 1 h at RT. Primary antibodies were diluted in blocking buffer (1% donkey serum) and incubated overnight at 4°C. After three 5 min washes with PBS, secondary fluorescently labeled antibodies were diluted in blocking buffer (1% donkey serum) and incubated for 1 h at RT. Coverslips were washed twice with 1x PBS and an additional two times with dH$_2$O and mounted with ProLong Gold antifade reagent (Thermo Fisher Scientific).

### Imaging LYTL tubules and microtubules in fixed cells
Cells were fixed with 4% PFA and 0.1% glutaraldehyde for 10 min at RT. After five washes with PBS, cells were incubated with 0.1% NaBH4 in PBS for 10 min at RT. After three washes, cells were incubated in 0.1% NaBH4 + 5% BSA in PBS for 1 h at 37°C. After three washes with PBS, cells were permeabilized with PBS/0.1% Triton for 10 min and blocked with 5% donkey serum for 1 h at RT. Primary antibodies were diluted in blocking buffer (1% donkey serum) and incubated overnight at 4°C. After three 5 min washes with PBS, secondary fluorescently labeled antibodies were diluted in blocking buffer (1% donkey serum) and incubated for 1 h at RT. Coverslips were washed twice with 1x PBS and an additional two times with dH$_2$O and mounted with Pro-Long Gold antifade reagent (Thermo Fisher Scientific).

## RILPL1:RAB10 structural binding model
The X-cap of RILPL1 was created using AlphaFold2 predictive modeling. The model of RAB10 was created based on the coordinates by Rai et al. (2016) (PDB ID: 5SZJ). Predictive interaction of RILPL1/RAB10 was created in AlphaFold2 Multimer. ChimeraX software (Pettersen et al., 2021) was used to visualize and render all structures.

## Coimmunoprecipitation
HEK293FT, HEK293T, and U2OS cells transfected with 3xflag-LRRK2 (or HaloTag-LRRK2) and mCherry-RILPL1 (or 2xmyc-RILPL1) plasmids were lysed in IP buffer (20 mM Tris-HCl [pH 7.5], 150 mM NaCl, 1 mM EDTA, 0.3% Triton X-100, 10% glycerol, and 1× Halt protease and phosphatase inhibitor cocktail) (Thermo Fisher Scientific) for 30 min on ice. Lysates were centrifuged at 4°C for 10 min at 20,000 g. The supernatant was incubated in RFP-Trap (or myc-Trap) magnetic beads

(ChromoTek) for 1 h at 4°C on a rotator. Beads were washed six times with IP wash buffer (20 mM Tris-HCl [pH 7.5], 300 mM NaCl, 1 mM EDTA, 0.1% Triton X-100, and 10% glycerol) and eluted in 2× loading buffer (Bio-Rad) with β-mercaptoethanol by boiling for 5 min at 95°C.

### SDS-PAGE and western blotting

Proteins were resolved on 4–20% Criterion TGX precast gels (Bio-Rad) and transferred to membranes by the semi-dry Trans-Blot Turbo transfer system (Bio-Rad). The membranes were blocked with Intercept Blocking Buffer (LI-COR) and then incubated for 1 h at RT or overnight at 4°C with the indicated primary antibody. The membranes were washed in TBST (three times for 5 min), followed by incubation for 1 h at RT with fluorescently conjugated goat anti-mouse, rat, or rabbit IR Dye 680 or 800 antibodies (LICOR). The blots were washed in TBST (three times for 5 min) and scanned on an ODYSSEY CLx (LICOR) and an Azure 500 (Azure). Quantitation of WBs was performed using Image Studio (LICOR) and Azure Spot Pro (Azure).

### Statistical analysis

Statistics and data analysis were performed using GraphPad Prism 10.0.1 (GraphPad). Statistical analysis for experiments with two treatment groups used Student's $t$ tests with Welch's correction for unequal variance. The F test was used to assess the equality of variances of the $t$ test. For more than two groups, we used one-way ANOVA or two-way ANOVA where there were two factors in the model. Tukey's post hoc test was used to determine statistical significance for individual comparisons in those cases where the underlying ANOVA was statistically significant and where all groups were compared; Dunnett's multiple comparison test was used where all groups were compared back with a single control group. For more than two groups and unequal SD's, Brown–Forsythe and Welch ANOVA was used. Unless otherwise stated, graphed data are presented as means ± SEM or violin plots. Data distribution was assumed to be normal, but this was not formally tested. Comparisons that were considered statistically significant are indicated; $*P < 0.05$; $**P < 0.01$; $***P < 0.001$; $****P < 0.0001$.

### Online supplemental material

Fig. S1 (related to Fig. 2) shows the characterization of the GFP-LRRK2, TMEM192-3xHA HEK293T cells, the Lyso-IP quality control, and the mass spectrometry data (DMSO vs LLOME). Fig. S2 (related to Fig. 2) shows the lipidomics profiling of RAB susbtrates, pRAB effectors, and dynein-related proteins. Fig. S3 complements Fig. 3, providing further information on RILPL1 as a protein recruited to lysosomes by LRRK2 kinase activity via pRAB proteins. Figs. S4 and S5 (related to Fig. 8) further explore the link between LYTL tubules and α-tubulin PTMs. Video 1 (related to Fig. 4) shows the perinuclear movement of LRRK2+ lysosomes. Video 2 (related to Fig. 6) depicts the dynamic behavior of RILPL1+ tubules. Video 3 comes from Fig. S4 A. Video 4 is from Fig. S4 E, and Video 5 is from Fig. S4 F. Video 6 is from Fig. 8 E. Table S1 is the raw data from the proteomics analysis in Fig. 2.

### Data availability

All the data needed to evaluate the conclusions in the paper are present in the paper and/or the supplementary materials. Additional data related to this paper may be requested from the authors.

## Acknowledgments

This research was supported by the Intramural Research Program of the National Institutes of Health, the National Institute on Aging (MRC), and by an National Institute of General Medical Sciences R35 grant (1R35GM157207; Luis Bonet-Ponce).

Author contributions: Luis Bonet-Ponce: conceptualization, data curation, formal analysis, funding acquisition, investigation, methodology, project administration, resources, supervision, visualization, and writing—original draft, review, and editing. Tsion Tegicho: investigation and methodology. Nuria Fernandez-Martinez: investigation. Irene A Rozenberg: investigation and visualization. Mia Ashriem: investigation. Alexandra G Beilina: formal analysis, investigation, and writing—review and editing. Jillian H Kluss: methodology, visualization, and writing—review and editing. Yan Li: investigation and methodology. Mark R Cookson: conceptualization, project administration, supervision, and writing—review and editing.

Disclosures: The authors declare no competing interests exist.

Submitted: 4 April 2024

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

Figure S1.   **Additional information regarding the LYSO-IP unbiased proteomics screening. (A and B)** HEK293T cells stably expressing GFP-LRRK2 and TMEM192-3xHA were seeded for 24 h and treated with DMSO or LLOME for 2 h. Cells were then fixed and stained using an anti HA antibody. **(C and D)** HEK293T cells stably expressing GFP-LRRK2 and TMEM192-3xHA were seeded for 24 h and treated with DMSO, LLOME (2 h), and LLOME+MLi2. Cells were then fixed and stained for HA and endogenous RAB10 (C) or endogenous JIP4 (D). **(E, F, and H)** HEK293T cells stably expressing GFP-LRRK2 and TMEM192-3xHA were seeded for 24 h and treated with DMSO, LLOME (2 h), and LLOME+MLi2. Cells were then subjected to LYSO-IP to immuno-isolate lysosomes. Isolated lysosomes were lysed, and immunoblots show the amount of different cellular compartments in the lysosomal fraction, including EE, ER, cytosol, Golgi, RE, and mitochondria (E), as well as different luminal and membranous lysosomal proteins (F). **(G)** Marker quantification confirming enrichment of lysosomal proteins in the lysosomal fraction. **(H)** Immunoblot from LYSO-IP of players involved in LYTL, including pT73-RAB10, total RAB10, LRRK2, and JIP4. **(I)** Volcano plot showing the proteins with enhanced recruitment to lysosomes (right side) and the proteins decreased on lysosomes (left side), when cells are treated with LLOME. Data are from four independent experiments. **(J)** Histogram shows protein levels measured by mass spectrometry of individual proteins on lysosomes under DMSO, LLOME, and LLOME+MLi2. Data are from four independent experiments. **(K)** Immunoblot from LYSO-IP experiment showing that RILPL1 recruitment to damaged lysosomes is LRRK2 kinase activity dependent. Source data are available for this figure: SourceData FS1.

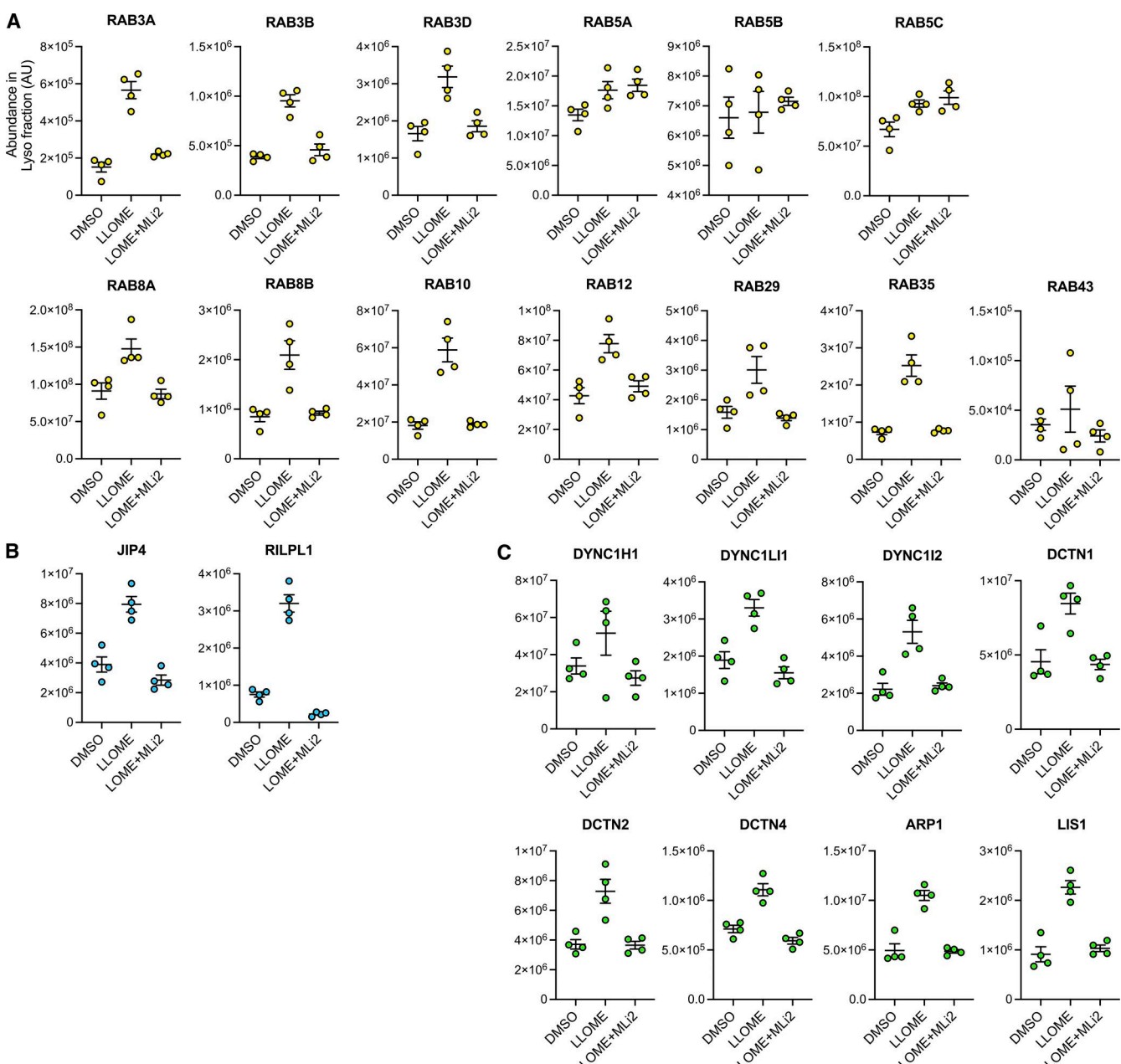

Figure S2. **Additional information regarding** Fig. 1. Histogram shows protein levels measured by mass spectrometry of individual proteins on lysosomes under DMSO, LLOME, and LLOME+MLi2. Data are from four independent experiments. **(A–C)** RAB substrates, (B) RHD proteins, and (C) dynein/dynactin subunits.

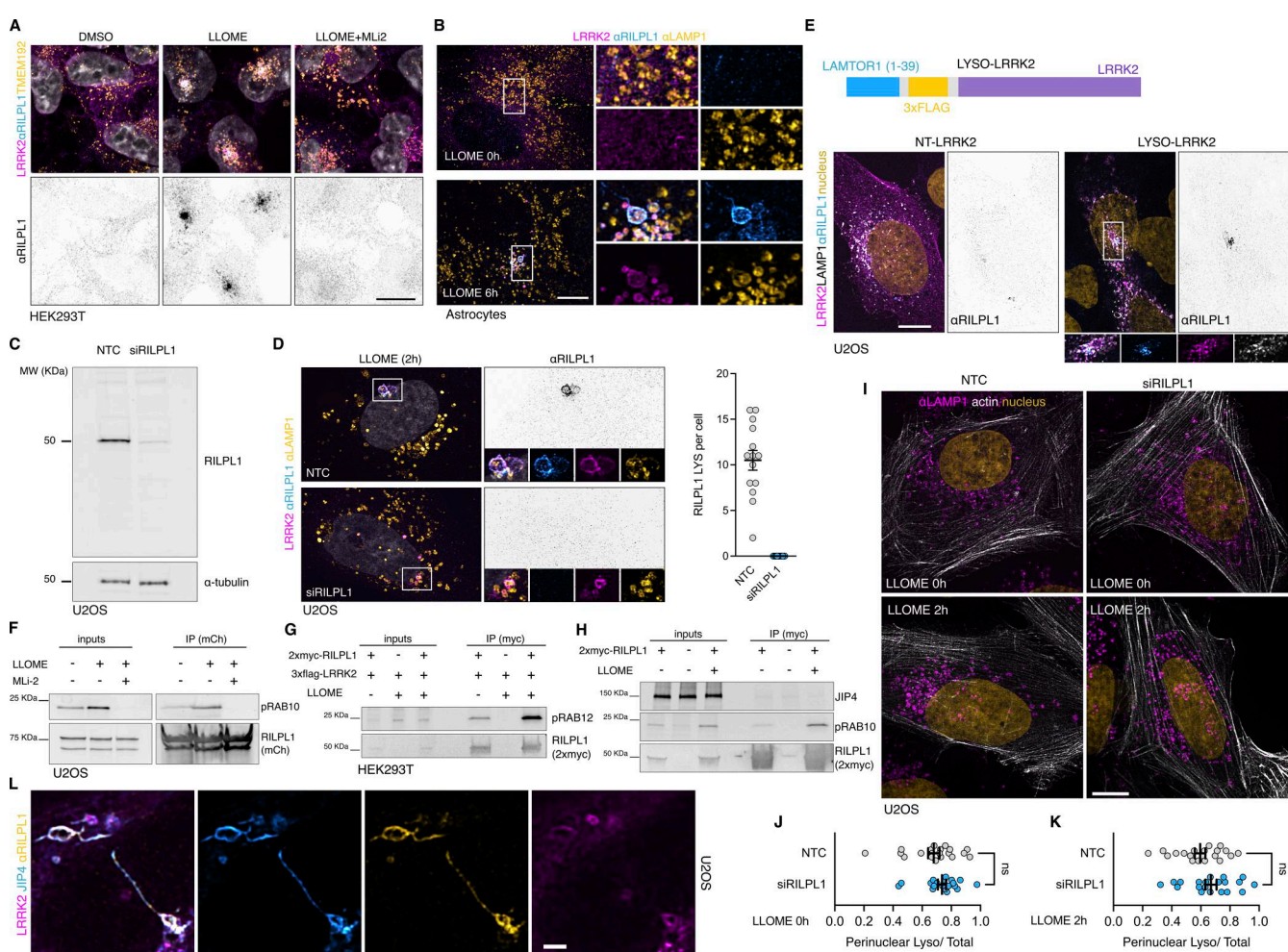

Figure S3. **Additional information regarding RILPL1 recruitment to lysosomes by LRRK2. (A)** HEK293T cells stably expressing GFP-LRRK2 and TMEM192-3xHA were seeded for 24 h and treated with DMSO, LLOME (2 h), and LLOME+MLi2. Cells were then fixed and stained for HA and endogenous RILPL1. **(B)** Mouse primary astrocytes were transfected with 3xflag-LRRK2 for 48 h. Cells were then treated or not with LLOME for 6 h, fixed, and stained for endogenous LAMP1 and RILPL1. **(C)** U2OS cells were treated with a nontargeting control (NTC) or siRILPL1 for 60 h. Cells were then lysed and blotted for endogenous RILPL1. **(D)** U2OS cells were treated with a NTC or siRILPL1 for 24 h. Cells were then transfected with 3xflag-LRRK2 for 36 h, treated with LLOME for 2 h, and fixed. Cells were stained for endogenous RILPL1 and LAMP1. Graph shows the number of RILPL1-positive lysosomes per cell in both conditions (*n* = 13–14 cells). **(E)** U2OS cells were transfected with LAMP1-HaloTag and 3xflag-LRRK2 (NT-LRRK2) or LAMTOR1(1–39)-3xflag-LRRK2 (LYSO-LRRK2) for 36 h. Cells were fixed and stained for endogenous RILPL1. **(F)** U2OS cells were transfected with HaloTag-LRRK2 and mCherry-RILPL1 for 36 h. Cells were then treated or not with LLOME (2 h) or pre-treated with MLi2. Lysates were subjected to immunoprecipitation with anti-RFP antibodies. **(G)** HEK293T cells were transfected with 3xflag-LRRK2 and 2xmyc-RILPL1 for 24 h. Then cells were treated or not with LLOME for 2 h. Lysates were subjected to immunoprecipitation with anti-myc antibodies and blotted against pS106-RAB12 and myc antibodies. **(H)** HEK293T cells were transfected with 3xflag-LRRK2 and 2xmyc-RILPL1 for 24 h. Then cells were treated or not with LLOME for 2 h. Lysates were subjected to immunoprecipitation with anti-myc antibodies and blotted against JIP4, pT73-RAB10, and myc antibodies. **(I)** U2OS cells were treated with a NTC or siRILPL1 for 24 h. Cells were then transfected with LifeAct-mNeonGreen for 36 h, treated or not with LLOME for 2 h, and fixed. Cells were then stained for endogenous LAMP1. **(J and K)** Graphs show the perinuclear ratio of lysosomes in the different conditions. Unpaired *t* test. Data are mean ± SEM (*n* = 20 cells). (L) U2OS cells transfected with HaloTag-LRRK2 and mNeonGreen-JIP4 for 36 h and treated with LLOME for 2 h before fixing. Cells were then stained for endogenous RILPL1. Scale bar = 10 μm. Source data are available for this figure: SourceData FS3.

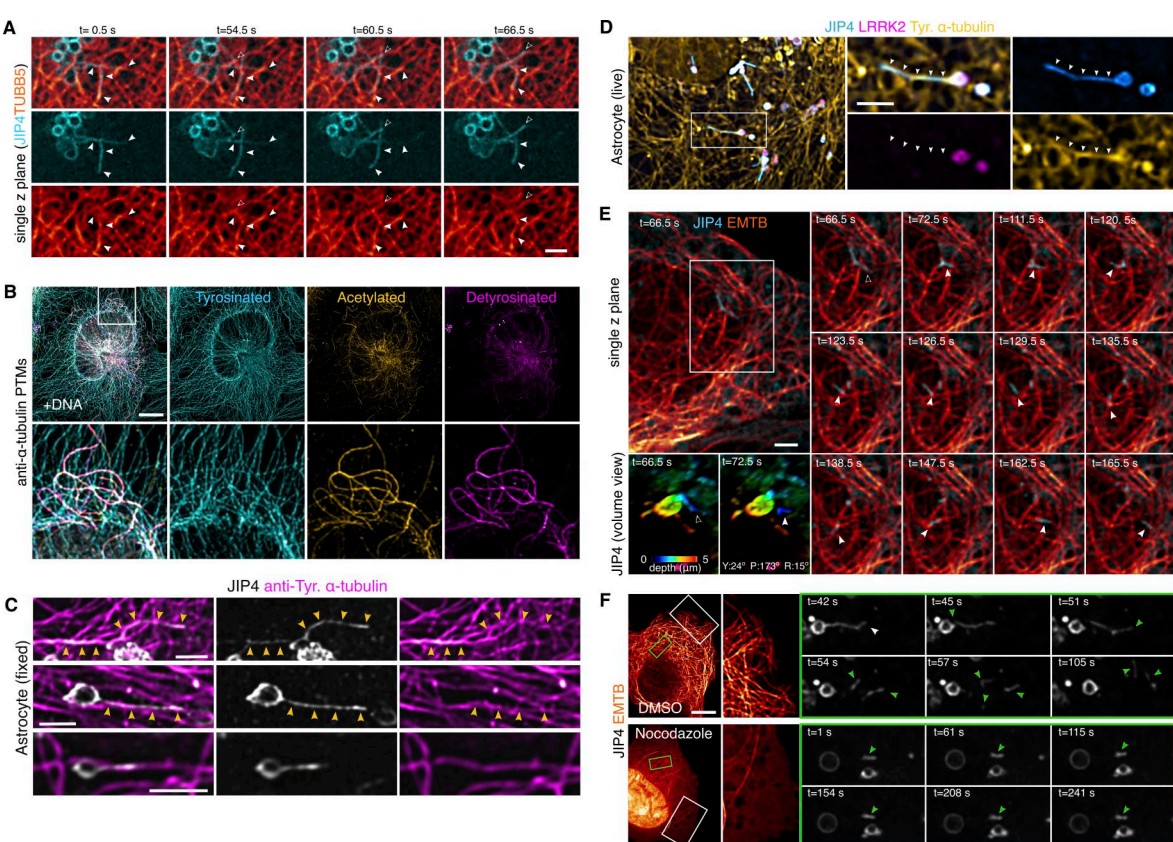

Figure S4. **Additional information regarding α-tubulin PTM contacts with LYTL tubules. (A)** U2OS cells were transfected with 3xflag-LRRK2, mNeonGreen-JIP4, and TUBB5-HaloTag. Cells were treated with LLOME (2 h) and observed under a confocal microscope. Time-lapse shows LYTL tubules associated with microtubules (arrowheads) and even a tubule budding alongside a microtubule (outlined arrowhead). **(B)** U2OS cells were fixed and stained for different α-tubulin PTMs: tyrosinated, acetylated, and detyrosinated. **(C)** Mouse primary astrocytes were transfected with 3xflag-LRRK2 and pDEST53-JIP4 for 48 h. Cells were treated with LLOME (6 h) and fixed. After fixation, cells were stained for GFP and tyrosinated α-tubulin. Yellow arrowheads indicate contact between LYTL tubules and tyrosinated microtubules. **(D)** Mouse primary astrocytes were transfected with HaloTag-LRRK2, mNeonGreen-JIP4, and TagRFP-T-A1aY1 (tyrosinated microtubules) for 48 h. Cells were treated with LLOME (6 h) and imaged live. Arrowheads indicate contact between LYTL tubules and tyrosinated microtubules. **(E)** U2OS cells were transfected with 3xflag-LRRK2, HaloTag-JIP4, and EMTB-mNeonGreen. Cells were treated with LLOME (2 h) and analyzed under a confocal microscope. Time-lapse shows a sorted tubule moving along microtubules on a single z plane. Volume view with depth code shows the tubule attached to a lysosome (t = 66.5 s) and the moment of fission (t = 72.5 s). **(F)** U2OS cells were transfected with 3xflag-LRRK2, HaloTag-JIP4, and EMTB-mNeonGreen. Cells were treated with LLOME (2 h) and with DMSO or nocodazole (10 µM, 30 min) and analyzed under a confocal microscope. Time-lapse shows the effect of nocodazole on the microtubule network and the dynamics of the LYTL-sorted materials. Scale bar (A and C–E) = 2 µm; (B and F) = 10 µm.

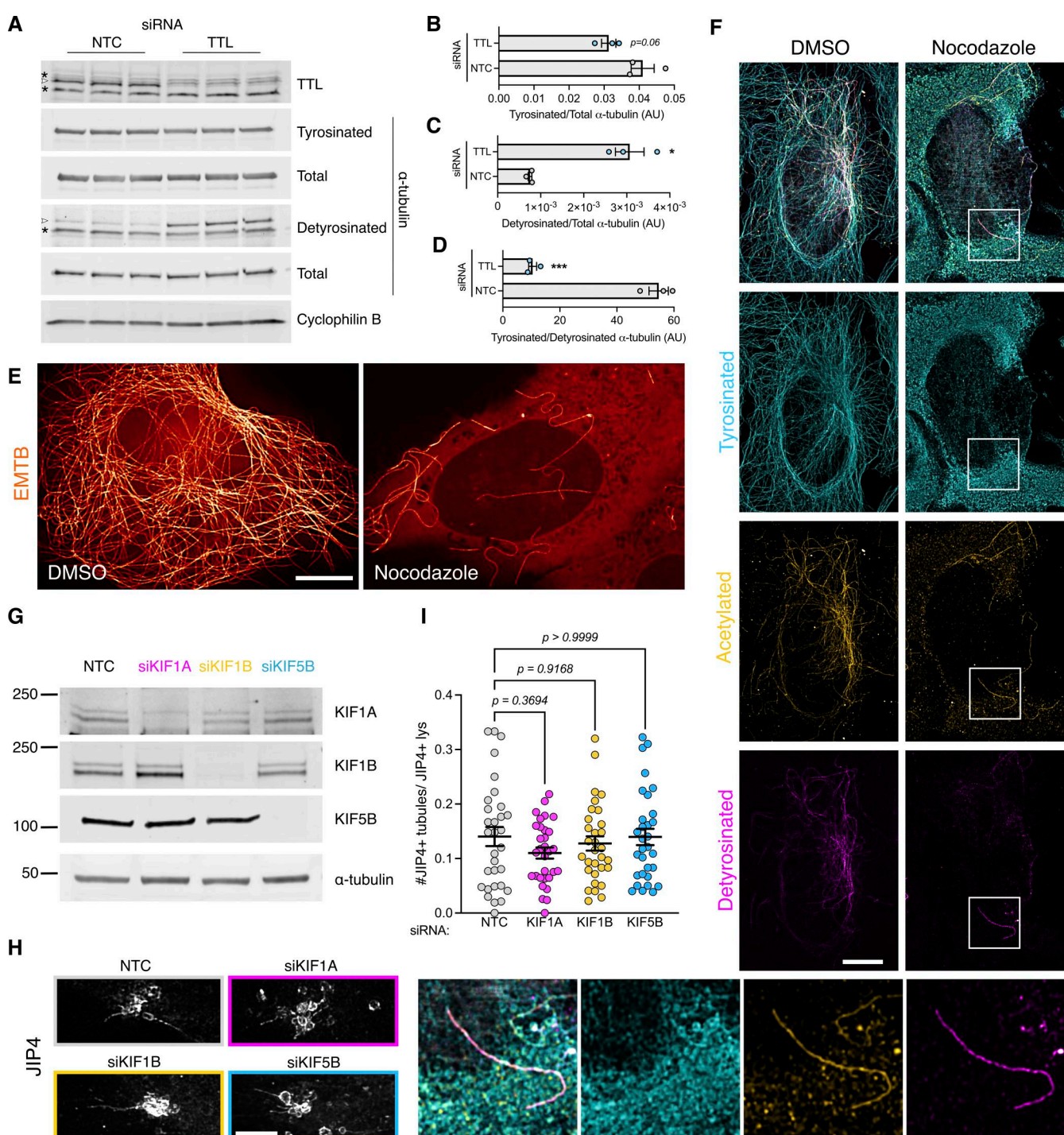

Figure S5.   **Additional information regarding tyrosinated α-tubulin and LYTL tubule elongation. (A)** U2OS cells were transfected with a nontargeting control (NTC) or TTL siRNA for 60 h. WB shows tyrosinated α-tubulin, detyrosinated α-tubulin, total α-tubulin, and cyclophilin B protein levels. **(B)** Histogram showing tyrosinated α-tubulin protein levels normalized to total α-tubulin. Unpaired $t$ test was used (P = 0.066, $n$ = 3). **(C)** Histogram showing detyrosinated α-tubulin protein levels normalized to total α-tubulin. Unpaired $t$ test with Welch's correction was used (P = 0.0185, $n$ = 3). **(D)** Histogram showing the normalized tyrosinated/detyrosinated α-tubulin ratio. Unpaired $t$ test was applied (P = 0.0003, $n$ = 3). Asterisks signal unspecific bands; real bands are shown in arrowheads. **(E)** U2OS cells were transfected with EMTB-mNeonGreen and treated with DMSO or nocodazole (10 µM, 30 min) and imaged live under a confocal microscope. **(E)** U2OS cells were treated with DMSO or nocodazole (10 µM, 30 min) and fixed. **(F)** U2OS cells treated with DMSO or nocodazole were fixed and stained for tyrosinated α-tubulin, detyrosinated α-tubulin, and acetylated α-tubulin. **(G and H)** U2OS cells were transfected with a NTC or KIF1A, KIF1B, and KIF5B siRNA for 24 h. Cells were then transfected with HaloTag-LRRK2 and mNeonGreen-JIP4 for 36 h and treated with LLOME for 2 h. Cells were then fixed and stained for endogenous LAMP2. **(I)** Graph depicts the #JIP4+ tubules/JIP4+ lysosomes ratio in all conditions. Brown–Forsythe and Welch ANOVA with Dunnett's. Data are mean ± SEM ($n$ = 31 cells). Scale bar = 10 µm. Source data are available for this figure: SourceData FS5.

Video 1. **U2OS cell transfected with HaloTag-LRRK2 for 36 h and treated with LLOME.** Maximum intensity projection. The cell was imaged for 120 min at 5 min per stack. The movie is played at 7 s per frame. Scale bar: 10 µm.

Video 2. **U2OS cells were transfected with 3xflag-LRRK2 and mBaoJin-RILPL1 for 36 h and treated with LLOME (2).** Maximum intensity projection. The cell was imaged for 3 min at 3 s per stack. The movie is played at 7 s per frame.

Video 3. **U2OS cells were transfected with 3xflag-LRRK2, mNeonGreen-JIP4, and TUBB5-HaloTag.** Subsequently, cells were treated with LLOME (2 h). The cell was imaged for 4 min at 3 s per stack. The movie is played at 4 s per frame. White arrowheads indicate LYTL tubules associated with microtubules, and outlined arrowheads show a tubule budding along a microtubule. Scale bar: 2 µm.

Video 4. **U2OS cells were transfected with 3xflag-LRRK2, HaloTag-JIP4, and EMTB-mNeonGreen.** Subsequently, cells were treated with LLOME (2 h). The cell was imaged for 4 min at 3 s per stack. The movie is played at 4 s per frame. White arrowheads indicate LYTL sorted material traveling on microtubules. Scale bar: 2 µm.

Video 5. **U2OS cells were transfected with 3xflag-LRRK2, HaloTag-JIP4, and EMTB-mNeonGreen.** Subsequently, cells were treated with LLOME (2 h) with (DMSO) or with nocodazole (10 µM, 30 min). Both cells were imaged for 4 min at 3 s per stack. Movies are played at 7 s per frame. White arrowheads indicate LYTL tubules, and outlined arrowheads show sorted material. Scale bar: 2 µm.

Video 6. **U2OS cells were transfected with 3xflag-LRRK2, mNeonGreen-JIP4, and TagRFP-A1aY.** Subsequently, cells were treated with LLOME (2 h). The cell was imaged for 105 s at 3 s per stack. Movies are played at 7 s per frame. White arrowheads indicate a sorted tubule moving along a tyrosinated microtubule. Scale bar: 2 µm.

**Provided online is Table S1. Table S1 shows label-free proteomic analysis of DMSO-, LLOME-, and LLOME+MLi2-treated cells in HEK293T stably expressing TMEM192-3xHA and GFP-LRRK2.**

