## [Peer Review File · The Journal of Cell Biology]

Opposing actions of JIP4 and RILPL1 provide antagonistic motor force to control lysosomal tubulation

Luis Bonet-Ponce, Tsion Tegicho, Nuria Fernandez-Martinez, Irene Rozenberg, Mia Ashriem, Alexandria Beilina, Jillian Kluss, Yan Li, and Mark Cookson

Corresponding Author(s): Luis Bonet-Ponce, The Ohio State University

Review Timeline:

Submission Date:	2024-04-04
Editorial Decision:	2024-04-25
Revision Received:	2025-05-22
Editorial Decision:	2025-06-09
Revision Received:	2025-08-01

Monitoring Editor: Hong Zhang

Scientific Editor: Andrea Marat

Transaction Report:

DOI: <https://doi.org/10.1083/jcb.202404018>

April 25, 2024

Re: JCB manuscript #202404018

Dr. Luis Bonet-Ponce
The Ohio State University
460 Medical Center Dr
425A
Columbus, Ohio 43065

Dear Dr. Bonet-Ponce,

Thank you for submitting your manuscript entitled "Opposing actions of JIP4 and RILPL1 provide antagonistic motor force to control lysosomal tubulation". The manuscript was assessed by expert reviewers, whose comments are appended to this letter. We invite you to submit a revision if you can address the reviewers' key concerns, as outlined here.

As you will see, the reviewers all appreciate that your study provides interesting new insight into the role of LRRK2 in regulating lysosomal damage. However, they have noted that additional evidence for the mechanistic role of RILPL1 in lysosomal tubulation is required to support your model and have provided constructive suggestions to this effect. Given the complexity of testing all Rabs identified in your screen, while we would welcome data to the effect, experiments with additional Rabs other than Rab10 are not required. Please however justify why Rab10 is the focus of the current work as noted by reviewer 1. You also do not need to examine potential kinesins given the complexities noted by reviewer 3. Otherwise, we expect you to address all of the remaining reviewer comments in your revised manuscript.

GENERAL GUIDELINES:

Text limits: Character count for an Article is < 40,000, not including spaces. Count includes title page, abstract, introduction, results, discussion, and acknowledgments. Count does not include materials and methods, figure legends, references, tables, or supplemental legends.

Figures: Articles may have up to 10 main text figures. Figures must be prepared according to the policies outlined in our Instructions to Authors, under Data Presentation, <https://jcb.rupress.org/site/misc/ifora.xhtml>. All figures in accepted manuscripts will be screened prior to publication.

*****IMPORTANT:** It is JCB policy that if requested, original data images must be made available. Failure to provide original images upon request will result in unavoidable delays in publication. Please ensure that you have access to all original microscopy and blot data images before submitting your revision. ***

Supplemental information: There are strict limits on the allowable amount of supplemental data. Articles may have up to 5 supplemental figures. Up to 10 supplemental videos or flash animations are allowed. A summary of all supplemental material should appear at the end of the Materials and methods section.

Please note that JCB now requires authors to submit Source Data used to generate figures containing gels and Western blots with all revised manuscripts. This Source Data consists of fully uncropped and unprocessed images for each gel/blot displayed in the main and supplemental figures. Since your paper includes cropped gel and/or blot images, please be sure to provide one Source Data file for each figure that contains gels and/or blots along with your revised manuscript files. File names for Source Data figures should be alphanumeric without any spaces or special characters (i.e., SourceDataF#, where F# refers to the associated main figure number or SourceDataFS# for those associated with Supplementary figures). The lanes of the gels/blots should be labeled as they are in the associated figure, the place where cropping was applied should be marked (with a box), and molecular weight/size standards should be labeled wherever possible.

The typical timeframe for revisions is three to four months. While most universities and institutes have reopened labs and

allowed researchers to begin working at nearly pre-pandemic levels, we at JCB realize that the lingering effects of the COVID-19 pandemic may still be impacting some aspects of your work, including the acquisition of equipment and reagents. Therefore, if you anticipate any difficulties in meeting this aforementioned revision time limit, please contact us and we can work with you to find an appropriate time frame for resubmission. Please note that papers are generally considered through only one revision cycle, so any revised manuscript will likely be either accepted or rejected.

Thank you for this interesting contribution to Journal of Cell Biology. You can contact us at the journal office with any questions at cellbio@rockefeller.edu.

Sincerely,

Hong Zhang, PhD
Monitoring Editor

Andrea L. Marat, PhD
Senior Scientific Editor

Journal of Cell Biology

Reviewer #1 (Comments to the Authors (Required)):

The manuscript shows RILPL1 as a downstream effector of LRRK2, recruited by Rab10 to damaged lysosomes. RILPL1 mediates lysosome positioning and tubulation (LYTL) by interacting with the dynein-dynactin complex. Mutations in LRRK2 are the most common genetic cause of Parkinson's disease, and cell responses to lysosome damage are important to cell survival. The mechanism of LYTL formation and its functions are significant in understanding PD pathology. Although the connection between LRRK2 and RILPL1 in lysosomes has been reported, this work demonstrates the molecular function of RILPL1 in regulation of lysosome positioning and LYTL, extending the understanding of the mechanism of LRRK2-regulated lysosome dynamics. The experiments were carefully designed, and the results were clearly presented.

My major comments include:

1. The authors show that RILPL1 regulates lysosome retrograde transport and LYTL tubule dynamics. However, the relationship between these two events was not addressed. Fig. 3F, H, I show that RILPL1 KD decreases JIP4 association to the lysosome, and the authors think the reason is "RILPL1 and JIP4 likely bind pRAB proteins using similar motifs but in an antagonistic manner" (Line 197). In the authors' previous report, lysosome positioning to the perinuclear area promotes p-Rab10, JIP4 lysosomal localization, and LYTL formation (Kluss et al., 2022, PNAS). Could this be a reason for decreased JIP4 in RILPL1 KD cells? It would be interesting to include RILPL1 and JIP4 in Fig 3E and examine RILPL1 and JIP4 in dynein inhibitor-treated cells.
2. Related to the above comment, Fig. 3F, H, I show that RILPL1 KD decreases JIP4 lysosomal association, and Fig. 4E shows tubular ratio (JIP4+tubules/JIP4+lysosomes) increased in the KD cells. These results seem controversial. Could the authors clarify the role of RILPL1 in the tubule formation?
3. In this study, the authors did not show JIP4 binding to kinesins or the defect of JIP4 deficiency-caused tubule antegrade growth, and JIP4 is known to be capable of binding to both kinesins and dynein-dynactin based on the published works. Thus, the authors assume JIP4 binds to kinesin in LYTL tubules and propose "Opposing actions of JIP4 and RILPL1 provide antagonistic motor force to control lysosomal tubulation" (the title), which sounds lacking support for JIP4's role.

Some minor comments:

1. The authors may want to justify why only rab10 was studied, since other rabs were found by MS.
2. Line 156 "The presence of active LRRK2...". What is "active LRRK2"?
3. The size of WB DLIC in Fig. 5D doesn't align with other blots.

Reviewer #2 (Comments to the Authors (Required)):

In this study, Bonet-Ponce L et al. investigated new players involved in LYTL (LYsosomal Tubulation/sorting driven by LRRK2)

using unbiased LRRK2 lysosomal proteomic analysis. The authors found that LRRK2 recruits RILP-like protein 1 (RILPL1) to ruptured lysosomes through pRABs. They propose that RILPL1 and JIP4, which they reported in previous work, interact with pRABs with similar mechanisms. They further validated that RILPL1 binds to p150Glued to facilitate the movement of lysosomal tubules to the minus end of microtubules, therefore clustering LRRK2-positive lysosomes towards nucleus. They suggest that JIP4 and RILPL1 act as opposing motor adaptor proteins to provide forces required for lysosomal tubulation induced by lysosome damage.

While this study revealed a potential role of RILPL1 in LRRK2-dependent lysosomal tubulation initiated by lysosome damage, the mechanistic evidence that establishes the LRRK2-pRABs-RILPL1/JIP4 is missing. It is not addressed how RILPL1 and JIP4 coordinately regulate lysosomal tubulation upon lysosome damage.

Major concerns

1. The LRRK2-lysosome proteome analysis identified 9 RABs that are recruited to damaged lysosomes by LRRK2, it is not understood how individual of these RABs play a role in lysosome tubulation upon lysosome damage. This should be examined experimentally.
2. The authors conclude that RILPL1 is recruited by pRABs by providing the coIP data that pRAB10 interacts with RILPL1. Do other pRABs identified in this study also interact with RILPL1? Because each RAB probably acts differently through multiple effectors in lysosome tubulation.
3. The experimental data are missing that pRABs mediate/promote LRRK2 kinase activity-dependent recruitment of RILPL1 to lysosomes. Does knockdown or overexpression of RAB dominant-negative forms disrupt lysosomal recruitment of RILPL1/JIP4 in the presence of lysosome damage?
4. RILPL1 and JIP4 have conserved R residues in the RHD2 motif, which are predicted to mediate their binding to the phosphorylated residues in pRABs (Figure 2E). The authors thus suggest that RILPL1 and JIP4 interact with RABs with similar mechanism. Do RILPL1 and JIP4 interact with RABs (i.e., pRAB10) competitively, or whether they form tripartite complex with pRABs (pRAB10)? The current data seem not sufficient to conclude that RILPL1 and JIP4 bind pRAB proteins using similar motifs in an antagonistic manner.
5. The authors showed that in control cells LRRK2-positive lysosomes cluster to the centrosome (Figure 3E). How do LRRK2-lysosomes behave dynamically in cells with perturbed RILPL1 or JIP4 levels, i.e., knockdown or over-expression of RILPL1/JIP4? This will provide additional evidence to determine if RILPL1 really clusters LRRK2-lysosomes to the centrosome.
6. Given that RILPL1 knockdown significantly decreased the presence of JIP4 on LRRK2-positive lysosomes (Figure 3F, H, I) and that loss of RILPL1 increased JIP4-positive tubules (Figure 4C, E), does the decrease of lysosomal JIP4 result from the increase in JIP4 tubules?
7. RILPL1 colocalizes with JIP4 on the lysosomal tubules (Figure 4B). The previous study of the authors has shown that perinuclear clustering promotes the ability of LRRK2 to recruit JIP4 and initiate LYTL. However, this study showed that RILPL1 knockdown increased JIP4-lysosomal tubules (Figure 4C, E). Thus, it is difficult to draw the conclusion that RILPL1 and JIP4 control lysosome tubulation in an antagonistic manner.
8. To conclude that RILPL1 and JIP4 are two players involved in LYTL tubulation, it is necessary to analyze the dynamics of LYTL tubules in RILPL1- and/or JIP4-deficient cells using lysosomal tubule markers other than RILPL1/JIP4.
9. Do RILPL1 and JIP4 have a role in tubulin tyrosination? This can be tested in cells with knockdown of these two proteins.
10. The authors suggested that lysosome tubules on damaged lysosomes likely transfer undegraded cargos (see discussion), can they provide evidence?

Other points

1. In Fig. 2D, the input protein levels are not compatible. It is difficult to assess whether the interaction of RILPL1 with pRAB10 is dependent on LRRK2 kinase activity.
2. Why is LRRK2 recruited to a subset but not all cellular lysosomes after LLOME treatment? Are these lysosomes ruptured (this can be examined with galectin 3)?
3. In Fig. 5E, the input proteins are not at similar levels so as to quantify p150Glued protein levels from the lysosomal fraction.
4. Why does RILPL1 exhibit non-single bands in western blots, for example, in Fig. S3F and Fig. S3G.
5. Several images need scale bars.

Reviewer #3 (Comments to the Authors (Required)):

In this manuscript, Bonet-Pence et al describe a novel dual microtubule motor mechanism underlying the process of lysosomal tubulation/sorting driven by LRRK2 (LYTL), which allows for the reformation of functional lysosomes following lysosomal damage. Using a proteomics approach and subsequent validation by immunofluorescence microscopy and live cell imaging with fluorescent protein fusions, they show that lysosomal damage induces the LRRK2 activity-dependent recruitment of the microtubule motor adaptors JIP4 and RILPL1 (as well as a variety of other proteins, most notably LRRK2 target Rab GTPases). They show convincingly that RILPL1 is necessary for the motility of damaged lysosomes to the pericentriolar region and paradoxically for the maximal recruitment of JIP4. They then show that both RILPL1 and JIP4 are both simultaneously detected on tubules that emanate from the LRRK2-containing lysosomes upon damage. Several data suggest that while JIP4 is

responsible for tubule extension along tyrosinated microtubules, RILPL1 - through binding to dynactin and hence linking to dynein - is responsible for tubule retraction, giving rise to dynamic tubules that may break up into new lysosomes.

The paper addresses an important and interesting question related to how cells compensate for lysosomal damage, and provides compelling, visually stunning, and well-quantified data to support the major conclusions. Most conclusions are supported by multiple approaches. The experiments appear to be very well done and analyzed, and the story is relatively complete. The only missing piece is the identity of the kinesin mediating JIP4-dependent tubule extension, but given the complexity of the kinesin family and the negative result obtained with the most obvious candidate, this could take a long time (and potential redundancy might further complicate the search). The paper makes a significant advance in understanding the mechanism behind LYTL, which the authors recently discovered and reported elsewhere. Thus, in principle, this paper would be of interest to readers of the Journal of Cell Biology.

There are a few concerns that need to be addressed for the paper to meet the standards of the Journal of Cell Biology. Most can be addressed by modifications to the text or the figures without addition of new data. The majority relate to over- or mis-interpretation of data for minor conclusions throughout the paper and lack of clarity in some of the language. There are also some strange conventions within the figures that should be rectified and improvements in labeling of the figures for their actual content. At most only a few controls and quantification of existing data should be added. The concerns are detailed below.

Major concerns:

1. Why do the authors think that LRRK2 and its substrates are recruited to only a subset of lysosomes after LLOME treatment (a 2-6 h treatment at the concentration indicated should damage most lysosomes in the cells)? Are these the only damaged ones (e.g. are they also the only ones labeled by GFP-Gal3)? Minimally, reference to a previous paper may be sufficient to answer this and the text should admit to only a fraction of lysosomes becoming positive.
2. In Fig. 2D, is LLOME treatment alone (without excess LRRK2 expression) sufficient to detect the interaction of mCh-RILPL1 with pRAB10? In U2OS cells, this interaction does not appear to be dependent on lysosomal damage by LLOME; this should be noted in the text, and if there is an increase, it should be quantified.
3. Fig. 2F is a nice structural model based on AlphaFold2 multimer, but it is not an actual structure of the pRAB10/RILPL1 complex. Thus, the text on lines 173-174 massively overstates and clouds the conclusion of the interaction of these two proteins. It must be stated in the text that this is a structural model based on AlphaFold predictions, and clarified that the model predicts that each monomer of an antiparallel RILPL1 RH2 domain dimer interacts with a pRAB10 via the R293-T73 interaction. The predicted structure does not "confirm" anything - it supports a model in which the interaction might occur by a similar mechanism as in RILPL2 binding to pRAB8A. Moreover, the image for the RILPL1-R293A-expressing cell in Figure 2G does not strongly support the model that RILPL1 is recruited via the R293A-dependent interaction with pRAB10. In this figure there is an accumulation of labeling for RILPL1 in the vicinity of the lysosomes, including some labeled by LRRK2. They are not as bloated as the lysosomes in RILPL1-WT expressing cells and do not appear to generate tubules, but the loss of actual recruitment is not obvious. A more convincing figure and/or quantification of the result would better support the conclusion.
4. The conclusion on line 197-198 "RILPL1 an JIP4 likely bind pRAB proteins using similar motifs but in an antagonistic manner" does not follow from the data shown in Fig. 3 and contradicts the data to come in Fig. 4. The data in Fig. 3 show that RILPL1 recruitment appears to be necessary for optimal JIP4 recruitment to LRRK2-positive lysosomes, and in Fig. 4 the data show JIP4 and RILPL1 associated on the same structures - thus they do not bind antagonistically (which most readers would interpret as competitively). Why do the authors draw this conclusion? Similarly, what do the authors mean on lines 208-209 when they say (referring to the very clear data in Fig. 4 that RILPL1 depletion enhances JIP4-dependent tubulation) "These results oppose the effect previously reported with JIP4"? How do they oppose the effect, and what exactly are the data that you refer to?
5. Regarding Figure 5 and the text describing it:
 - In describing the interaction between RILPL1 and p150Glued in Figure 5D, the authors neglect to note in the text that the interaction was stronger in cells treated with LLOME, supporting their model. This should be noted if reproducible, and if not, a more representative blot should be shown.
 - The authors did not, as stated in the text on line 232, reproduce the recruitment of dynein subunits, LIS1, and four dynactin complex members in Fig. 5E-F - only p150Glued.
 - On line 236, the data do not show that RILPL1 is totally responsible for dynactin recruitment, only partially responsible since the reduction in p150Glued content is small upon RILPL1 knockdown.
 - Finally, the authors cannot conclude from the data shown that RILPL1 interacts directly with p150Glued, since this was drawn from a pull-down from cell lysates; other dynactin subunits might also be present and mediate the interaction. Thus, the conclusion on lines 243-244 should be tempered to "through its interaction with dynactin". This also needs to be modified in the Discussion (lines 323-330).
6. Text line 267-268 - what is meant by "sorted material"? The figure only shows mNeonGreen-JIP4. There is no evidence of any kind of sorting here other than the material that is left behind (LAMP1 and LRRK2). Do the authors mean a tubule that has broken off from a lysosome? If the authors wish to define this as "sorted", then they need to do so and explain why they use that

term.

7. In the discussion, the authors may consider how simultaneous recruitment of dynein and kinesin by RILPL1 and JIP4, respectively, might lead to breakage of the tubular structures into vesicular structures as they have previously reported.

Minor concerns:

8. On line 155 of the text describing recruitment of RILPL1 to live cells, the authors should indicate that the figure shows recruitment of mCherry-RILPL1 to LAMP1-NeonGreen-labeled lysosomes by live cell imaging (i.e., not endogenous RILPL1).

9. Since the authors use multiple cell types throughout the paper, it would be helpful to readers if they indicated on the figures themselves which cell type is used in that panel/ figure.

10. The text on lines 200-202 is confusing as written - as is it sounds like the lysosomes are contradictorily positive and negative for LRRK2 and LAMP1. I believe what the authors mean is "...identified RILPL1 tubules that are negative for LRRK2 and LAMP1 but that emanate from LRRK2-positive lysosomes".

11. Although they are defined clearly in the Materials and Methods, it would be helpful to briefly define quantitative terms such as "perinuclear ratio" (Figure 2), "lysosomal ratio" (Figure 3), and "tubular ratio" (Figures 4 and 6) in the text and/or figure legend because they are not obvious terms and one cannot understand the data without an idea of what they mean. Alternatively, on the axes and in the text "perinuclear ratio" can be replaced by "perinuclear/ total lys.", "tubular ratio" can be replaced by "# JIP4+ tubules/ JIP4+ lys", and "lysosomal ratio" by "Lysosomal fraction" (this measurement is not a ratio).

12. The y-axis in Figure 6B should not say "%".

13. Please note that the word "data" is the plural of datum and should be used with a plural verb.

We thank the reviewers for their insightful comments and their effort on improving our manuscript. In this document, all reviewer's comments are in **bold** and our responses are in plain font.

Reviewer #1:

My major comments include:

1. The authors show that RILPL1 regulates lysosome retrograde transport and LYTL tubule dynamics. However, the relationship between these two events was not addressed. Fig. 3F, H, I show that RILPL1 KD decreases JIP4 association to the lysosome, and the authors think the reason is "RILPL1 and JIP4 likely bind pRAB proteins using similar motifs but in an antagonistic manner" (Line 197). In the authors' previous report, lysosome positioning to the perinuclear area promotes p-Rab10, JIP4 lysosomal localization, and LYTL formation (Kluss et al., 2022, PNAS). Could this be a reason for decreased JIP4 in RILPL1 KD cells? It would be interesting to include RILPL1 and JIP4 in Fig 3E and examine RILPL1 and JIP4 in dynein inhibitor-treated cells.

We apologize for the unclear explanation in the original text and appreciate the reviewer's feedback. We agree that the loss of perinuclear clustering caused by RILPL1 knockdown likely reduces JIP4 recruitment to lysosomes, as previously shown in our *Kluss et al., 2022 (PNAS)* study. We have revised the sentence to read: "Our results are consistent with our prior observations (Kluss et al., 2022a) that perinuclear clustering of LRRK2-positive lysosomes is important to recruit JIP4." Additionally, we have included two new experiments: (1) time-lapse imaging of RILPL1 knockdown cells showing a loss of LRRK2+ lysosome clustering (Fig. 4E), and (2) reduced clustering of RILPL1 and LRRK2+ lysosomes following siDYNC1H1 knockdown (Fig. 4F–G).

2. Related to the above comment, Fig. 3F, H, I show that RILPL1 KD decreases JIP4 lysosomal association, and Fig. 4E shows tubular ratio (JIP4+tubules/JIP4+lysosomes) increased in the KD cells. These results seem controversial. Could the authors clarify the role of RILPL1 in the tubule formation?

As the reviewer noted, our model proposes two distinct functions for RILPL1: (1) it mediates clustering of LRRK2+ lysosomes toward the microtubule minus-end, and (2) it drives retraction of LYTL tubules in the same direction. We believe these are independent roles. In RILPL1 knockdown cells, reduced lysosomal clustering leads to decreased JIP4 recruitment. Meanwhile, the loss of RILPL1-driven retraction results in longer and more abundant tubules.

3. In this study, the authors did not show JIP4 binding to kinesins or the defect of JIP4 deficiency-caused tubule antegrade growth, and JIP4 is known to be capable of binding to both kinesins and dynein-dynactin based on the published works. Thus, the authors assume JIP4 binds to kinesin in LYTL tubules and propose "Opposing actions of JIP4 and RILPL1 provide antagonistic motor force to control lysosomal tubulation" (the title), which sounds lacking support for JIP4's role.

In our previous paper (Bonet-Ponce et al., 2020, *Sci Adv*), we showed that JIP4 knockdown reduces lysosomal tubulation. At the time, we used RAB10 as a tubulation marker because it was the only other known protein present in LYTL tubules, though at lower levels than JIP4. Since RILPL1 is now recognized as a more reliable LYTL tubule marker, comparable to JIP4, we repeated the experiment using tagged RILPL1. As shown in Fig. 6G–I, JIP4 knockdown cells display shorter tubules and a lower tubulation ratio, confirming that RILPL1 and JIP4 have opposing roles in regulating lysosomal tubulation

Some minor comments:

1. The authors may want to justify why only rab10 was studied, since other rabs were found by MS.

RAB10 was found to interact physically with JIP4 in our previous paper (Bonet-Ponce et al., 2020. *Sci Adv*). Moreover, T73-RAB10 is one of only two reliable antibodies for LRRK2 RAB substrates available. The other one being S106-RAB12 (which we confirm its interaction with RILPL1 in Fig. S3G). We included the following sentence “We focused on RAB10 for two main reasons: i) is the pRAB with the best characterized antibody, and ii) we previously showed that RAB10 binds and recruits JIP4 to lysosomes (Bonet-Ponce et al., 2020).”

2. Line 156 "The presence of active LRRK2...". What is "active LRRK2"?

By “active LRRK2” we refer to LRRK2 being kinase active.

3. The size of WB DLIC in Fig. 5D doesn't align with other blots.

We apologize for our mistake. It has been corrected.

Reviewer #2:

In this study, Bonet-Ponce L et al. investigated new players involved in LYTL (LYsosomal Tubulation/sorting driven by LRRK2) using unbiased LRRK2 lysosomal proteomic analysis. The authors found that LRRK2 recruits RILP-like protein 1 (RILPL1) to ruptured lysosomes through pRABs. They propose that RILPL1 and JIP4, which they reported in previous work, interact with pRABs with similar mechanisms. They further validated that RILPL1 binds to p150Glued to facilitate the movement of lysosomal tubules to the minus end of microtubules, therefore clustering LRRK2-positive lysosomes towards nucleus. They suggest that JIP4 and RILPL1 act as opposing motor adaptor proteins to provide forces required for lysosomal tubulation induced by lysosome damage.

While this study revealed a potential role of RILPL1 in LRRK2-dependent lysosomal tubulation initiated by lysosome damage, the mechanistic evidence that establishes the LRRK2-pRABs-RILPL1/JIP4 is missing. It is not addressed how RILPL1 and JIP4 coordinately regulate lysosomal tubulation upon lysosome damage.

Major concerns

1. The LRRK2-lysosome proteome analysis identified 9 RABs that are recruited to damaged lysosomes by LRRK2, it is not understood how individual of these RABs play a role in lysosome tubulation upon lysosome damage. This should be examined experimentally.

The reviewer raises an important point. Unfortunately, only two reliable antibodies are currently available for the nine RABs identified—pRAB10 and pRAB12—which limits our ability to study the other RAB substrates. As noted above and in the revised manuscript, RAB10 is a strong candidate because we previously demonstrated its physical interaction with JIP4 and its role in recruiting JIP4 to LRRK2+ lysosomes (*Bonet-Ponce et al., 2020, Sci Adv*). We have added the following sentence to the manuscript: “We focused on RAB10 for two main reasons: i) is the pRAB with the best characterized antibody, and ii) we previously showed that RAB10 binds and recruits JIP4 to lysosomes (Bonet-Ponce et al., 2020).”

2. The authors conclude that RILPL1 is recruited by pRABs by providing the coIP data that pRAB10 interacts with RILPL1. Do other pRABs identified in this study also interact with RILPL1? Because each RAB probably acts differently through multiple effectors in lysosome tubulation.

We have included a new experiment assessing the interaction with the other reliable antibody (S106-RAB12; Fig. S3G). We observe a clear binding between RILPL1 and pRAB12 that is already strong in the DMSO condition and increases slightly after LLOME addition.

3. The experimental data are missing that pRABs mediate/promote LRRK2 kinase activity-dependent recruitment of RILPL1 to lysosomes. Does knockdown or overexpression of RAB dominant-negative forms disrupt lysosomal recruitment of RILPL1/JIP4 in the presence of lysosome damage?

The reviewer is correct, and we have added an experiment comparing wild-type RAB10 with the phospho-null mutant T73A-RAB10 to assess recruitment of endogenous RILPL1 to lysosomes. As expected, cells expressing the phospho-null RAB10 show a strong and significant reduction in RILPL1 recruitment to LRRK2+ lysosomes (Fig. 3H).

4. RILPL1 and JIP4 have conserved R residues in the RHD2 motif, which are predicted to mediate their binding to the phosphorylated residues in pRABs (Figure 2E). The authors thus suggest that RILPL1 and JIP4 interact with RABs with similar mechanism. Do RILPL1 and JIP4 interact with RABs (i.e., pRAB10) competitively, or whether they form tripartite complex with pRABs (pRAB10)? The current data seem not sufficient to conclude that RILPL1 and JIP4 bind pRAB proteins using similar motifs in an antagonistic manner.

Our new data seem to suggest that RILPL1 and JIP4 don't bind to pRAB10 in a competitive manner, as JIP4 overexpression enhances the RILPL1 binding to pRAB10 (Fig. 3I). We were unable to detect direct interaction between RILPL1 and JIP4 (Fig. S3H), suggesting that both effectors don't form a tripartite interaction with the pRABs but are not competing for their binding. We have added the following

sentence to the text: “It is worth noting that a competition between both effector proteins is unlikely, as JIP4 overexpression increases the RILPL1: pRAB10 interaction, suggesting a further stabilization of the complex. Although a direct interaction between RILPL1 and JIP4 was not detected, the relationship between JIP4 and RILPL1 is worth exploring in future endeavors.”

5. The authors showed that in control cells LRRK2-positive lysosomes cluster to the centrosome (Figure 3E). How do LRRK2-lysosomes behave dynamically in cells with perturbed RILPL1 or JIP4 levels, i.e., knockdown or over-expression of RILPL1/JIP4? This will provide additional evidence to determine if RILPL1 really clusters LRRK2-lysosomes to the centrosome.

As per the reviewer’s suggestion, we have studied the dynamic behavior of LRRK2+ lysosomes in RILPL1 KD cells. Fig. 4E, shows lack of clustering of LRRK2+ lysosomes in cells treated with siRILPL1, confirming the role of RILPL1 in clustering LRRK2+ lysosomes to the centrosome.

6. Given that RILPL1 knockdown significantly decreased the presence of JIP4 on LRRK2-positive lysosomes (Figure 3F, H, I) and that loss of RILPL1 increased JIP4-positive tubules (Figure 4C, E), does the decrease of lysosomal JIP4 result from the increase in JIP4 tubules?

As we stated above, our model proposes two distinct functions for RILPL1: (1) it mediates clustering of LRRK2+ lysosomes toward the microtubule minus-end, and (2) it drives retraction of LYTL tubules in the same direction. We believe these are independent roles. In RILPL1 knockdown cells, reduced lysosomal clustering leads to decreased JIP4 recruitment. Meanwhile, the loss of RILPL1-driven retraction results in longer and more abundant tubules.

7. RILPL1 colocalizes with JIP4 on the lysosomal tubules(Figure 4B). The previous study of the authors has shown that perinuclear clustering promotes the ability of LRRK2 to recruit JIP4 and initiate LYTL. However, this study showed that RILPL1 knockdown increased JIP4-lysosomal tubules(Figure 4C, E). Thus, it is difficult to draw the conclusion that RILPL1 and JIP4 control lysosome tubulation in an antagonistic manner.

We have knockdown JIP4 and analyze tubulation using RILPL1. As shown in Fig. 6G–I, JIP4 knockdown cells display shorter tubules and a lower tubulation ratio, confirming that RILPL1 and JIP4 have opposing roles in regulating lysosomal tubulation

8. To conclude that RILPL1 and JIP4 are two players involved in LYTL tubulation, it is necessary to analyze the dynamics of LYTL tubules in RILPL1- and/or JIP4-deficient cells using lysosomal tubule markers other than RILPL1/JIP4.

The reviewer raises an important point: the need for a marker of LYTL tubules that is independent of pRABs or their effectors—ideally, a typical lysosomal membrane protein. Unfortunately, we have not identified such a marker. So far, we have tested TMEM192, LAMP1, LAMP2, LIMP2, ARL8B, and LAMTOR4

(Bonet-Ponce et al., 2020, *Sci Adv*; Bonet-Ponce et al., 2022, *MBoC*). Currently, JIP4 and RILPL1 are the only available reliable markers study LYTL. We have discussed this limitation in the discussion: “It is also notable the need to find other LYTL tubule markers independent of JIP4 and RILPL1 to be able to identify all possible tubulation events.”

9. Do RILPL1 and JIP4 have a role in tubulin tyrosination? This can be tested in cells with knockdown of these two proteins.

We thank the reviewer for the great suggestion. We have done the experiment (Fig. 8I-J), and observed a significant reduction in tubulin tyrosination only after RILPL1 knockdown.

10. The authors suggested that lysosome tubules on damaged lysosomes likely transfer undegraded cargos (see discussion), can they provide evidence?

We have shown that LYTL vesicles establish contact with healthy lysosomes after being released from the tubules. Our current hypothesis is that the contact creates an environment for cargo transfer. However, as we currently lack proper evidence of cargo transfer (i.e. cargo identification), we have softened this sentence by “LYTL originates from compromised lysosomes and could serve as a mechanism for transferring undegraded material from dysfunctional lysosomes to active ones. However, we are still working to identify the cargo carried by LYTL vesicles.”

Other points

1. In Fig. 2D, the input protein levels are not compatible. It is difficult to assess whether the interaction of RILPL1 with pRAB10 is dependent on LRRK2 kinase activity.

The input levels on pRAB10 are different given the difference in treatment (LLOME) or LRRK2 expression. Both are known to affect pRAB10 levels.

2. Why is LRRK2 recruited to a subset but not all cellular lysosomes after LLOME treatment? Are these lysosomes ruptured (this can be examined with galectin 3)?

We apologize for the lack of information that might be accessible to readers not familiar with our previous published work. In order to properly introduce the recruitment of LRRK2 to membrane damaged lysosomes, we have created an additional figure (Fig. 1), where we show the recruitment of LRRK2 to lysosomes in cells treated with LLOME (Fig. 1A-C). We have also analyzed LRRK2 colocalization with GAL3 (Fig. 1D). In short, most of LRRK2+ lysosomes are also labeled with GAL3 (~70%), demonstrating the recruitment to membrane damaged lysosomes. However, only ~14% of GAL3+ lysosomes are decorated with LRRK2. This illustrates the complex recruitment of LRRK2 to damaged lysosomes which complete mechanism remains unclear.

3. In Fig. 5E, the input proteins are not at similar levels so as to quantify p150Glued protein levels from the lysosomal fraction.

We understand this point, however, the p150Glued levels were analyzed from the IP'd fraction normalizing by the levels of LAMP1. We have done this experiment three independent times, and we are certain of its validity.

4. Why does RILPL1 exhibit non-single bands in western blots, for example, in Fig. S3F and Fig. S3G.

It seems to happen only in U2OS cells and tagged with mCherry; it might be related to protein processing.

5. Several images need scale bars.

We have corrected this issue.

Reviewer #3 (Comments to the Authors (Required)):

In this manuscript, Bonet-Pence et al describe a novel dual microtubule motor mechanism underlying the process of lysosomal tubulation/sorting driven by LRRK2 (LYTL), which allows for the reformation of functional lysosomes following lysosomal damage. Using a proteomics approach and subsequent validation by immunofluorescence microscopy and live cell imaging with fluorescent protein fusions, they show that lysosomal damage induces the LRRK2 activity-dependent recruitment of the microtubule motor adaptors JIP4 and RILPL1 (as well as a variety of other proteins, most notably LRRK2 target Rab GTPases). They show convincingly that RILPL1 is necessary for the motility of damaged lysosomes to the pericentriolar region and paradoxically for the maximal recruitment of JIP4. They then show that both RILPL1 and JIP4 are both simultaneously detected on tubules that emanate from the LRRK2-containing lysosomes upon damage. Several data suggest that while JIP4 is responsible for tubule extension along tyrosinated microtubules, RILPL1 - through binding to dynactin and hence linking to dynein - is responsible for tubule retraction, giving rise to dynamic tubules that may break up into new lysosomes.

The paper addresses an important and interesting question related to how cells compensate for lysosomal damage, and provides compelling, visually stunning, and well-quantified data to support the major conclusions. Most conclusions are supported by multiple approaches. The experiments appear to be very well done and analyzed, and the story is relatively complete. The only missing piece is the identity of the kinesin mediating JIP4-dependent tubule extension, but given the complexity of the kinesin family and the negative result obtained with the most obvious candidate, this could take a long time (and potential redundancy might further complicate the search). The paper makes a

significant advance in understanding the mechanism behind LYTL, which the authors recently discovered and reported elsewhere. Thus, in principle, this paper would be of interest to readers of the Journal of Cell Biology.

We have added two more kinesins previously related to lysosomes (KIF1A and KIF1B) that also fail to reduce tubulation upon its knockdown (Fig. S5G-I). We are currently expanding the search with promising candidates, that will be the focus of a follow up study.

There are a few concerns that need to be addressed for the paper to meet the standards of the Journal of Cell Biology. Most can be addressed by modifications to the text or the figures without addition of new data. The majority relate to over- or mis-interpretation of data for minor conclusions throughout the paper and lack of clarity in some of the language. There are also some strange conventions within the figures that should be rectified and improvements in labeling of the figures for their actual content. At most only a few controls and quantification of existing data should be added. The concerns are detailed below.

Major concerns:

1. Why do the authors think that LRRK2 and its substrates are recruited to only a subset of lysosomes after LLOME treatment (a 2-6 h treatment at the concentration indicated should damage most lysosomes in the cells)? Are these the only damaged ones (e.g. are they also the only ones labeled by GFP-Gal3)? Minimally, reference to a previous paper may be sufficient to answer this and the text should admit to only a fraction of lysosomes becoming positive.

As we show in the new Fig. 1 (and in Bonet-Ponce et al., 2020), these are not the only damaged lysosomes in the cell. Indeed, even though most LRRK2+ lysosomes are also labeled with GFP-Gal3 (~70%), the vast majority of Gal3+ lysosomes lack LRRK2. This suggest that membrane damage on itself is not the only stimulus required to recruit LRRK2 to lysosomes and that, a very tight and highly control (and mostly unknown to date) mechanism is in place.

2. In Fig. 2D, is LLOME treatment alone (without excess LRRK2 expression) sufficient to detect the interaction of mCh-RILPL1 with pRAB10? In U2OS cells, this interaction does not appear to be dependent on lysosomal damage by LLOME; this should be noted in the text, and if there is an increase, it should be quantified.

U2OS cells or HEK293T do not express detectable levels of endogenous LRRK2, which is why we use exogenous expression systems. According to the reviewer's suggestion, we have removed Fig. S3F.

3. Fig. 2F is a nice structural model based on AlphaFold2 multimer, but it is not an actual structure of the pRAB10/RILPL1 complex. Thus, the text on lines 173-174 massively overstates and clouds the conclusion of the interaction of these two proteins. It must be stated in the text that this is a structural

model based on AlphaFold predictions, and clarified that the model predicts that each monomer of an antiparallel RILPL1 RH2 domain dimer interacts with a pRAB10 via the R293-T73 interaction. The predicted structure does not "confirm" anything - it supports a model in which the interaction might occur by a similar mechanism as in RILPL2 binding to pRAB8A. Moreover, the image for the RILPL1-R293A-expressing cell in Figure 2G does not strongly support the model that RILPL1 is recruited via the R293A-dependent interaction with pRAB10. In this figure there is an accumulation of labeling for RILPL1 in the vicinity of the lysosomes, including some labeled by LRRK2. They are not as bloated as the lysosomes in RILPL1-WT expressing cells and do not appear to generate tubules, but the loss of actual recruitment is not obvious. A more convincing figure and/or quantification of the result would better support the conclusion.

We agree with the reviewer and apologize for poor writing. The sentence has now been changed to "Our structural modelling using AlphaFold predicts that each monomer of an antiparallel RILPL1 RHD2 domain dimer interacts with pThr.73-RAB10 via Arg.293 forming a tetramer (Fig. 3F)". Moreover, we have changed the picture in Fig. 3G showing a more convincing lack of RILPL1 recruitment in cells expressing the RILPL1-R293A mutant.

4. The conclusion on line 197-198 "RILPL1 and JIP4 likely bind pRAB proteins using similar motifs but in an antagonistic manner" does not follow from the data shown in Fig. 3 and contradicts the data to come in Fig. 4. The data in Fig. 3 show that RILPL1 recruitment appears to be necessary for optimal JIP4 recruitment to LRRK2-positive lysosomes, and in Fig. 4 the data show JIP4 and RILPL1 associated on the same structures - thus they do not bind antagonistically (which most readers would interpret as competitively). Why do the authors draw this conclusion? Similarly, what do the authors mean on lines 208-209 when they say (referring to the very clear data in Fig. 4 that RILPL1 depletion enhances JIP4-dependent tubulation) "These results oppose the effect previously reported with JIP4"? How do they oppose the effect, and what exactly are the data that you refer to?

In our previous paper (Bonet-Ponce et al., 2020, *Sci Adv*), we showed that JIP4 knockdown reduces lysosomal tubulation. At the time, we used RAB10 as a tubulation marker because it was the only other known protein present in LYTL tubules, though at lower levels than JIP4. Since RILPL1 is now recognized as a reliable LYTL tubule marker, comparable to JIP4, we repeated the experiment using tagged RILPL1. As shown in Fig. 6G-I, JIP4 knockdown cells display shorter tubules and a lower tubulation ratio, confirming that RILPL1 and JIP4 have opposing roles in regulating lysosomal tubulation

5. Regarding Figure 5 and the text describing it:

- In describing the interaction between RILPL1 and p150Glued in Figure 5D, the authors neglect to note in the text that the interaction was stronger in cells treated with LLOME, supporting their model. This should be noted if reproducible, and if not, a more representative blot should be shown.
- The authors did not, as stated in the text on line 232, reproduce the recruitment of dynein subunits, LIS1, and four dynactin complex members in Fig. 5E-F - only p150Glued.

- On line 236, the data do not show that RILPL1 is totally responsible for dynactin recruitment, only partially responsible since the reduction in p150^{Glued} content is small upon RILPL1 knockdown.
- Finally, the authors cannot conclude from the data shown that RILPL1 interacts directly with p150^{Glued}, since this was drawn from a pull-down from cell lysates; other dynactin subunits might also be present and mediate the interaction. Thus, the conclusion on lines 243-244 should be tempered to "through its interaction with dynactin". This also needs to be modified in the Discussion (lines 323-330).

- We apologize for not including the stronger interaction in LLOME treated cells. We have now included the following sentence in the Results section "RILPL1: p150^{Glued} binding is increased upon LLOME addition, supporting our model."

- The reviewer is correct. The new sentence reads: "We validated the recruitment of p150^{Glued} using western blotting (Fig. 7E,F), demonstrating that LRRK2 kinase activity is required to recruit p150^{Glued} to damaged lysosomes in a similar pattern to RILPL1."

- We agree with the reviewer and apologize for the lack of oversight. We have added "partially" to the sentence: "RILPL1 knockdown significantly reduces the amount of p150^{Glued} on lysosomes in LLOME-treated cells (Fig. 7G,H), suggesting that RILPL1 is partially responsible for the recruitment of dynactin to damaged lysosomes (Fig. 7I)."

- The reviewer is correct. We have changed the text accordingly.

6. Text line 267-268 - what is meant by "sorted material"? The figure only shows mNeonGreen-JIP4. There is no evidence of any kind of sorting here other than the material that is left behind (LAMP1 and LRRK2). Do the authors mean a tubule that has broken off from a lysosome? If the authors wish to define this as "sorted", then they need to do so and explain why they use that term.

The reviewer is correct as no cargo sorting has been proven. We did mean it as a tubule broken off a lysosome and have changed to "fissioned tubule".

7. In the discussion, the authors may consider how simultaneous recruitment of dynein and kinesin by RILPL1 and JIP4, respectively, might lead to breakage of the tubular structures into vesicular structures as they have previously reported.

We thank the reviewer for such an interesting idea. We have now included the following sentence in the Discussion section: "The lack of RILPL1 leads to a decrease in tubule retraction and fission. Given the differential recruitment of motors by RILPL1 and JIP4 respectively, the combined retraction and elongation force might create a more favorable environment for tubule fission into moving vesicles."

Minor concerns:

8. On line 155 of the text describing recruitment of RILPL1 to live cells, the authors should indicate that the figure shows recruitment of mCherry-RILPL1 to LAMP1-NeonGreen-labeled lysosomes by live cell imaging (i.e., not endogenous RILPL1).

We have modified the text, that now reads: "RILPL1 recruitment to LRRK2-positive lysosomes was also observed in live cells (Fig. 3C) using exogenous expression of mCherry-RILPL1 and LAMP1-mNeonGreen."

9. Since the authors use multiple cell types throughout the paper, it would be helpful to readers if they indicated on the figures themselves which cell type is used in that panel/ figure.

We have included all cell types used in the Figures.

10. The text on lines 200-202 is confusing as written - as is it sounds like the lysosomes are contradictorily positive and negative for LRRK2 and LAMP1. I believe what the authors mean is "...identified RILPL1 tubules that are negative for LRRK2 and LAMP1 but that emanate from LRRK2-positive lysosomes".

We appreciate the reviewer's suggestion and have modified the text accordingly.

11. Although they are defined clearly in the Materials and Methods, it would be helpful to briefly define quantitative terms such as "perinuclear ratio" (Figure 2), "lysosomal ratio" (Figure 3), and "tubular ratio" (Figures 4 and 6) in the text and/or figure legend because they are not obvious terms and one cannot understand the data without an idea of what they mean. Alternatively, on the axes and in the text "perinuclear ratio" can be replaced by "perinuclear/ total lys.", "tubular ratio" can be replaced by "# JIP4+ tubules/ JIP4+ lys", and "lysosomal ratio" by "Lysosomal fraction" (this measurement is not a ratio).

We have changed the axes titles according to the reviewer's suggestions.

12. The y-axis in Figure 6B should not say "%".

We have corrected the Figure (Fig. 7B)

13. Please note that the word "data" is the plural of datum and should be used with a plural verb.

We thank the reviewer for pointing that out and have corrected the text accordingly.

June 9, 2025

RE: JCB Manuscript #202404018R

Luis Bonet-Ponce
The Ohio State University

Dear Dr. Bonet-Ponce:

Thank you for submitting your revised manuscript entitled "Opposing actions of JIP4 and RILPL1 provide antagonistic motor force to control lysosomal tubulation". We would be happy to publish your paper in JCB pending final revisions necessary to meet our formatting guidelines (see details below).

A. MANUSCRIPT ORGANIZATION AND FORMATTING:

1) Text limits: Character count for Articles is < 40,000, not including spaces. Count includes abstract, introduction, results, discussion, and acknowledgments. Count does not include title page, figure legends, materials and methods, references, tables, or supplemental legends.

2) Figures limits: Articles may have up to 10 main text figures.

3) Figure formatting: * Scale bars must be present on all microscopy images, including inset magnifications (you may alternatively indicate the inset diameter). Molecular weight or nucleic acid size markers must be included on all gel electrophoresis. Aspect ratios of images may not be altered.

* We appreciate that for most blots you have provided quantifications, however as noted by reviewer #2 please consider whether an alternative exposure may better depict the results.

4) Statistical analysis: Error bars on graphic representations of numerical data must be clearly described in the figure legend. The number of independent data points (n) represented in a graph must be indicated in the legend. Statistical methods should be explained in full in the materials and methods. For figures presenting pooled data the statistical measure should be defined in the figure legends. Please also be sure to indicate the statistical tests used in each of your experiments (either in the figure legend itself or in a separate methods section) as well as the parameters of the test (for example, if you ran a t-test, please indicate if it was one- or two-sided, etc.). Also, if you used parametric tests, please indicate if the data distribution was tested for normality (and if so, how). If not, you must state something to the effect that "Data distribution was assumed to be normal but this was not formally tested."

5) Abstract and title: The abstract should be no longer than 160 words and should communicate the significance of the paper for a general audience. The title should be less than 100 characters including spaces. Make the title concise but accessible to a general readership.

6) Materials and methods: Should be comprehensive and not simply reference a previous publication for details on how an experiment was performed. Please provide full descriptions in the text for readers who may not have access to referenced manuscripts.

7) All antibodies, cell lines, animals, and tools used in the manuscript should be described in full, including accession numbers for materials available in a public repository such as the Resource Identification Portal. Please be sure to provide the sequences for all of your primers/oligos and RNAi constructs in the materials and methods. You must also indicate in the methods the source, species, and catalog numbers (where appropriate) for all of your antibodies. Please also indicate the acquisition and quantification methods for immunoblotting/western blots.

8) Microscope image acquisition: The following information must be provided about the acquisition and processing of images:

- a. Make and model of microscope
- b. Type, magnification, and numerical aperture of the objective lenses
- c. Temperature
- d. Imaging medium
- e. Fluorochromes

- f. Camera make and model
 - g. Acquisition software
 - h. Any software used for image processing subsequent to data acquisition. Please include details and types of operations involved (e.g., type of deconvolution, 3D reconstitutions, surface or volume rendering, gamma adjustments, etc.).
- 9) References: There is no limit to the number of references cited in a manuscript. References should be cited parenthetically in the text by author and year of publication. Abbreviate the names of journals according to PubMed.
- 10) Supplemental materials: There are strict limits on the allowable amount of supplemental data. Articles may have up to 5 supplemental figures. Please also note that tables, like figures, should be provided as individual, editable files. A summary of all supplemental material should appear at the end of the Materials and methods section.
- 11) eTOC summary: A ~40-50-word summary that describes the context and significance of the findings for a general readership should be included on the title page. The statement should be written in the present tense and refer to the work in the third person.
- 12) Conflict of interest statement: JCB requires inclusion of a statement in the acknowledgements regarding competing financial interests. If no competing financial interests exist, please include the following statement: "The authors declare no competing financial interests." If competing interests are declared, please follow your statement of these competing interests with the following statement: "The authors declare no further competing financial interests."
- 13) ORCID IDs: ORCID IDs are unique identifiers allowing researchers to create a record of their various scholarly contributions in a single place. Please note that ORCID IDs are now **required** for all authors. At resubmission of your final files, please be sure to provide your ORCID ID and those of all co-authors.
- 14) A separate author contribution section following the Acknowledgments. All authors should be mentioned and designated by their full names. We encourage use of the CRediT nomenclature.

Please note that JCB now requires authors to submit Source Data used to generate figures containing gels and Western blots with all revised manuscripts. This Source Data consists of fully uncropped and unprocessed images for each gel/blot displayed in the main and supplemental figures. For assays performed using capillary electrophoresis and/or immunoassay-based detection, authors should instead provide the electropherogram graph(s) for each experiment, plotting fluorescence/chemiluminescence intensity vs. molecular weight/size. Please be sure to provide one Source Data file for each figure gels, blots, and/or capillary electrophoresis assays along with your revised manuscript files. File names for Source Data figures should be alphanumeric without any spaces or special characters (i.e., SourceDataF#, where F# refers to the associated main figure number or SourceDataFS# for those associated with Supplementary figures). For traditional gels and blots, the lanes of the gels/blots should be labeled as they are in the associated figure, the place where cropping was applied should be marked (with a box), and molecular weight/size standards should be labeled wherever possible. For capillary electrophoresis assays, each trace in the graph should be color-coded and labeled to indicate which protein, gene, or sample is being measured (please try to avoid red/green combinations to accommodate our color-blind readers).

Journal of Cell Biology now requires a data availability statement for all research article submissions. These statements will be published in the article directly above the Acknowledgments. The statement should address all data underlying the research presented in the manuscript. Please visit the JCB instructions for authors for guidelines and examples of statements at (<https://rupress.org/jcb/pages/editorial-policies#data-availability-statement>).

B. FINAL FILES:

Thank you for your attention to these final processing requirements. Please revise and format the manuscript and upload materials within 7 days. If you need an extension for whatever reason, please let us know and we can work with you to determine a suitable revision period.

Thank you for this interesting contribution, we look forward to publishing your paper in Journal of Cell Biology.

Sincerely,

Hong Zhang, PhD
Monitoring Editor

Andrea L. Marat, PhD
Deputy Editor

Journal of Cell Biology

Reviewer #1 (Comments to the Authors (Required)):

The authors have addressed all my questions satisfactorily.

Reviewer #2 (Comments to the Authors (Required)):

The authors have addressed most of my questions and the manuscript is significantly improved after revision. I support the publication of the revised paper; however, the following points need to be improved.

1. Fig. 3I, the authors are suggested to provide western blots with longer exposure time to support the conclusion that JIP4 increases the interaction of RILPL1 with pRAB10.
2. Likewise, protein bands in western blots in Fig. 7E (RILPL1 and p150Glued bands), Fig. 8I (Tyr. α -tubulin and RILPL1 bands), Fig. S3G (pRAB12 band), Fig. S5A (Tyr. α -tubulin band) are not strong enough to support the conclusions. The authors are suggested to provide blots with stronger signals.